# Irradiance and nutrient-dependent effects on photosynthetic electron transport in Arctic phytoplankton: A comparison of two chlorophyll fluorescence-based approaches to derive primary photochemistry

**Yayla Sezginer**[1]*, **David J. Suggett**[2], **Robert W. Izett**[1], **Philippe D. Tortell**[1,3]

**1** Department of Earth, Oceans, and Atmospheric Science, University of British Columbia, Vancouver, British Columbia, Canada, **2** Climate Change Cluster, University of Technology Sydney, Sydney, Australia, **3** Department of Botany, University of British Columbia, Vancouver, Canada

* ysezginer@eoas.ubc.ca

**Data Availability Statement:** Salinity and temperature data collected by Amundsen Science

## Abstract

We employed Fast Repetition Rate fluorometry for high-resolution mapping of marine phytoplankton photophysiology and primary photochemistry in the Lancaster Sound and Barrow Strait regions of the Canadian Arctic Archipelago in the summer of 2019. Continuous shipboard analysis of chlorophyll *a* variable fluorescence demonstrated relatively low photochemical efficiency over most of the cruise-track, with the exception of localized regions within Barrow Strait, where there was increased vertical mixing and proximity to land-based nutrient sources. Along the full transect, we observed strong non-photochemical quenching of chlorophyll fluorescence, with relaxation times longer than the 5-minute period used for dark acclimation. Such long-term quenching effects complicate continuous underway acquisition of fluorescence amplitude-based estimates of photosynthetic electron transport rates, which rely on dark acclimation of samples. As an alternative, we employed a new algorithm to derive electron transport rates based on analysis of fluorescence relaxation kinetics, which does not require dark acclimation. Direct comparison of kinetics- and amplitude-based electron transport rate measurements demonstrated that kinetic-based estimates were, on average, 2-fold higher than amplitude-based values. The magnitude of decoupling between the two electron transport rate estimates increased in association with photophysiological diagnostics of nutrient stress. Discrepancies between electron transport rate estimates likely resulted from the use of different photophysiological parameters to derive the kinetics- and amplitude-based algorithms, and choice of numerical model used to fit variable fluorescence curves and analyze fluorescence kinetics under actinic light. Our results highlight environmental and methodological influences on fluorescence-based photochemistry estimates, and prompt discussion of best-practices for future underway fluorescence-based efforts to monitor phytoplankton photosynthesis.

are available in the Polar Data Catalogue (doi: 10.5884/12715). Surface photosynthetically active radiation data are available in the Polar Data Catalogue (doi: 10.5884/12518), alongside additional meteorological data provided by the Amundsen Science group of U. Laval. CTD station conductivity, temperature, and salinity data provided by the Amundsen Science group of U. Laval are available in the Polar Data Catalogue (doi: 10.5884/12713). Underway oxygen saturation and ΔO2/Ar data are available in the Polar Data Catalogue (doi: 10.5884/13242). All Fast Repetition Rate Fluorometry measurements of phytoplankton photophysiology are available in the Polar Data Catalogue (doi: 10.5884/13254). The Canadian Cryospheric Information Network reference number for this dataset is 13254, and will be trackable using this number once published. A doi will be provided as soon as available. River data collected by the Canadian GEOTRACES program is available from the PANGEA database (https://doi.org/10.1594/PANGAEA.908497).

**Funding:** This work was supported by ArcticNet, a Network of Centres of Excellence of Canada and by the Natural Sciences and Engineering Research Council of Canada (NSERC) grants awarded to PT (https://www.nserc-crsng.gc.ca). The funders had no role in study design, data collection, decision to publish, or preparation of the manuscript.

**Competing interests:** The authors have declared that no competing interests exist.

## Introduction

Phytoplankton productivity in polar marine waters is constrained by nutrient and light availability, which fluctuate dramatically across seasonal cycles and shorter time and space scales [1]. In late summer, when sea ice cover is at a minimum and the mixed layer is shallow and highly stratified, phytoplankton are exposed to high solar irradiance and low nutrient concentrations [2]. Under these conditions, the growth and photosynthetic efficiency of Arctic phytoplankton becomes nitrogen-limited [3–5]. However, localized regions of elevated productivity can persist where various processes transport nutrients into the mixed layer, including upwelling, tidal mixing, and freshwater input from rivers and glaciers [6–8]. Modelling studies suggest that Barrow Strait in the Canadian Arctic Archipelago is one such productivity hotspot, with strong tidal currents and shallow sills driving vertical mixing in a region where Pacific and Atlantic-derived water masses converge [9, 10]. Additionally, Barrow Strait receives glacial and land-derived nutrients from the Cornwallis and Devon Island rivers [11, 12]. Rapid climate change in the Arctic is expected to have complex effects on these nutrient delivery mechanisms through the intensification of coastal erosion [13], increasing river inputs [14] and reduced vertical mixing due to intensifying stratification [15, 16]. At present, it is unclear how phytoplankton productivity will respond to these anticipated perturbations.

Assessing phytoplankton productivity in physically-dynamic marine waters requires high spatial resolution measurements that cannot be obtained from traditional discrete bottle incubation methods, such as $^{14}C$ uptake experiments. For this reason, oceanographic field studies have increasingly employed continuous sampling of surface water properties using variable chlorophyll $a$ (Chl$a$) fluorescence from Fast Repetition Rate Fluorometry (FRRf) and other related methods to rapidly and autonomously assess phytoplankton photophysiology and primary photochemistry as a proxy for primary productivity (e.g. [17–21]). Such variable fluorescence techniques rely on the inverse relationship between Chl$a$ fluorescence and photochemistry. These processes, along with heat dissipation, comprise the three energy dissipation pathways for light energy absorbed within Photosystem II (PSII) [22]. Fast Repetition Rate Fluorometry operates by supplying rapid excitation light pulses to progressively saturate the photosynthetic pathway and simultaneously induce a measurable Chl$a$ fluorescence response—often referred to as a fluorescence transient [23]. Analysis of Chl$a$ fluorescence transients provides information on the photochemical efficiency and functional absorption cross section of PSII, as well as estimates of the turnover rate of photosynthetic electron transport chain molecules [24, 25]. Fast Repetition Rate fluorescence transients can be obtained nearly instantaneously, offering opportunities for very high temporal and spatial resolution measurements.

Primary photochemistry is typically estimated from variable Chl$a$ fluorescence measurements by calculating the rate of photosynthetic electron transport out of PSII (ETR$_{PSII}$). There are several algorithms that may be applied to derive ETR$_{PSII}$ [17, 26–30]. Each algorithm relies on the same principles of light harvesting and photosynthetic electron transport, but arrive at ETR$_{PSII}$ estimates using slightly different, but theoretically equivalent, combinations of photophysiological metrics. As a result, different algorithms confer different field-sampling advantages and challenges (see [31]). The so-called 'amplitude' based approach (abbreviated ETR$_a$; sometimes also referred to as the sigma-algorithm), calculates ETR$_{PSII}$ as the product of photosynthetically active radiation (PAR), the functional absorption cross section of PSII in the dark-acclimated state ($\sigma_{PSII}$), and the photochemical efficiency of PSII normalized by the dark-acclimated maximum photochemical efficiency of PSII [27]. This approach reduces uncertainty in ETR$_{PSII}$ estimates by using $\sigma_{PSII}$ measurements made in the dark-acclimated state, which are subject to less noise than $\sigma_{PSII}$' measurements made in the light [31]. As a result,

$ETR_a$ is a favorable approach in low biomass waters where low signal-noise measurements represent a considerable challenge.

To achieve dark acclimation for $ETR_a$ measurements, phytoplankton samples are kept in darkness or very low light to relax non-photochemical quenching processes (NPQ), which upregulate heat dissipation of absorbed light energy, thereby reducing photochemistry and fluorescence yields [32]. Optimal dark acclimation times vary between phytoplankton species and depend to a large extent on the environmental history of samples [33], making it challenging to design widely applicable field protocols for high resolution data acquisition. In practice, applied dark-acclimation periods range from 5–30 minutes (e.g. [34, 35]). Many FRRf field deployments have focused on discrete sample analysis, applying extended dark acclimation periods to ensure samples reach a short-term steady state condition [25, 36–38]. Such discrete sample analysis enables standardized measurements and characterization of light-dependent physiological responses, at the cost of significantly reduced spatial and temporal measurement resolution. In contrast, continuous underway flow-through data acquisition yields high-resolution, real-time measurements of phytoplankton photophysiology, but creates uncertainty in the light exposure history of phytoplankton transiting through a ship's seawater supply lines. As a result, samples analyzed in continuous mode are neither fully representative of in-situ photophysiology or fully dark-acclimated states, and thus incompatible with the $ETR_a$ algorithm, which requires both dark- and light-acclimated measurements.

Recently, a new fluorescence approach has been developed to derive $ETR_{PSII}$ based on the turnover rate of the primary electron acceptor molecule ($Q_a$) within the photosynthetic electron transport chain [30]. In this 'kinetic' approach, $Q_a$ turnover rates are derived from analysis of fluorescence relaxation time-constants, resulting in the term $ETR_k$. The derivation of $ETR_k$ does not depend on dark-acclimated measurements, and can thus significantly increase the frequency of ship-board $ETR_{PSII}$ measurements. The kinetic fluorescence approach for $ETR_k$ was originally developed and implemented in mini-Fluorescence Induction-Relaxation (mini-FIRe) instruments [30], which use similar data acquisition protocols, but a different numerical approach for fluorescence relaxation analysis than FRRf. Gorbunov et al. [30] observed greater coherence between growth rates of laboratory cultures and FIRe-derived $ETR_k$ compared to alternative $ETR_{PSII}$ estimates, suggesting a strong potential for $ETR_k$ to quantify in-situ primary photochemistry. To our knowledge, there have been very few direct comparisons of FRRf-derived $ETR_k$ and $ETR_a$ estimates for natural phytoplankton assemblages.

In this article, we examine the relationship between $ETR_k$ and $ETR_a$ estimates across a range of hydrographic regimes in the Canadian Arctic Ocean. We employed a hybrid approach to data collection along a ship-track through Lancaster Sound and Barrow Strait, combining semi-continuous flow-through measurements with light response curves on static samples. Continuous sampling enabled us to obtain high-resolution measurements and examine the effects of light history and nutrient status on the physiology of Arctic phytoplankton assemblages. Data from rapid light-response curves allowed direct comparison of FRRf-derived $ETR_k$ and $ETR_a$ estimates. Our results demonstrate residual light-dependent NPQ effects on dark (low light) sample measurements, and a decoupling of $ETR_k$ and $ETR_a$ under conditions of phytoplankton photophysiological stress. We relate the spatial patterns in our observations to regional and fine-scale patterns in hydrography and nutrient supply in the eastern Canadian Arctic Archipelago, and discuss the potential effects of different data analysis approaches on $ETR_{PSII}$-based primary photochemistry estimates. Results from our work will inform future ship-based deployments of FRRf and related techniques to understand spatial patterns in phytoplankton productivity.

## Materials and methods

### Underway sampling

Arctic Ocean samples and hydrographic data were collected aboard the CCGS Amundsen from August 10–15, 2019, within the eastern region of Lancaster Sound and Barrow Strait, in the waters surrounding Devon and Cornwallis Islands. Access to these sampling regions was granted by the Nunavut Research Institute Scientific Research License (**0501119R-M**). All FRRf measurements were obtained with a LIFT (Light Induced Fluorescence Transient)-FRR fluorometer (LIFT-FRRf; Soliense Inc.). Water samples were delivered to the LIFT-FRRf sampling cuvette using the ship's seawater line as a primary supply (BC-4C-MD pump, March MFG Inc, nominal flow rate $\sim$20 L min$^{-1}$), combined with two secondary peristaltic pumps. The first pump (Masterflex L/S, model 7518–10), was used to create a continuous sampling loop ($\sim$200 mL min$^{-1}$) that was connected via t-fitting to a custom-built peristaltic pump actuated by the LIFT-FRRf software. The FRRf-actuated pump enabled precise synchronization of the sample handling and fluorescence measurements, allowing us to employ a semi-continuous sampling strategy of alternating fluorescence transients and light response curve measurements (details below).

In parallel with FRRf measurements, in-situ Chl*a* fluorescence, surface water salinity and temperature were measured using a flow-through Seabird thermosalinograph system (SBE 38), equipped with a WETStar fluorometer (WET Labs). The underway Chl*a* fluorescence signal was calibrated against discrete samples collected from surface Niskin bottles to approximate the along-track Chl*a* biomass (mg m$^{-3}$). Surface PAR measurements were obtained from an QCR-22000 Biospherical Instruments probe mounted above the ship's super-structure. Biological oxygen saturation, $\Delta O_2/Ar$, was measured as a metric of net community production using a Hiden Analytical quadrupole membrane inlet mass spectrometer (MIMS; HAL20) following the approaches outlined by Tortell et al. [39, 40]. These gas measurements were made continuously on seawater obtained from the same underway lines that supplied the FRRf system. Briefly, seawater was circulated at a constant flow rate past the mass spectrometer's inlet cuvette consisting of a 0.18 mm thick silicone membrane. Measurements of the mass-to-charge ratios at 32 ($O_2$) and 40 (Ar) atomic mass units were obtained at approximately 20 s. intervals. Air standards, consisting of filtered seawater ($<0.2$ $\mu$m) incubated at ambient sea surface temperature and gently bubbled using an aquarium air pump, were run periodically by automatically switching the inflow water source every 45–90 minutes. For both seawater and air standard measurements, the inflowing water passed through a 6-m heat exchange coil immersed in a constant-temperature 4°C water bath before passing the MIMS inlet. The seawater and air standard $O_2/Ar$ ratios ($[O_2/Ar]_{sw}$ and $[O_2/Ar]_{std}$, respectively) were used to derive underway $\Delta O_2/Ar$ by linearly interpolating between air standard measurements.

In addition to continuous surface water sampling, water column hydrographic profiles were examined using CTD casts to a depth of 200m at 16 stations (Fig 1). Mixed layer depths were derived from a density difference criterion of 0.125 kg m$^{-3}$ from surface water values. The CTD and underway surface salinity, temperature, Chl*a*, and PAR data were provided by the Amundsen Science group of Université Laval [42–44].

### LIFT-FRRf sampling protocols & parameter retrieval

Chlorophyll *a* fluorescence transients were obtained using a Single Turnover (ST) flash protocol and fit to the biophysical model of Kolber et al. [23] to derive photosynthetic parameters under dark acclimation and under actinic light, where the latter is denoted with the' notation. A summary of parameters measured and definitions is given in Table 1. Specifically, we

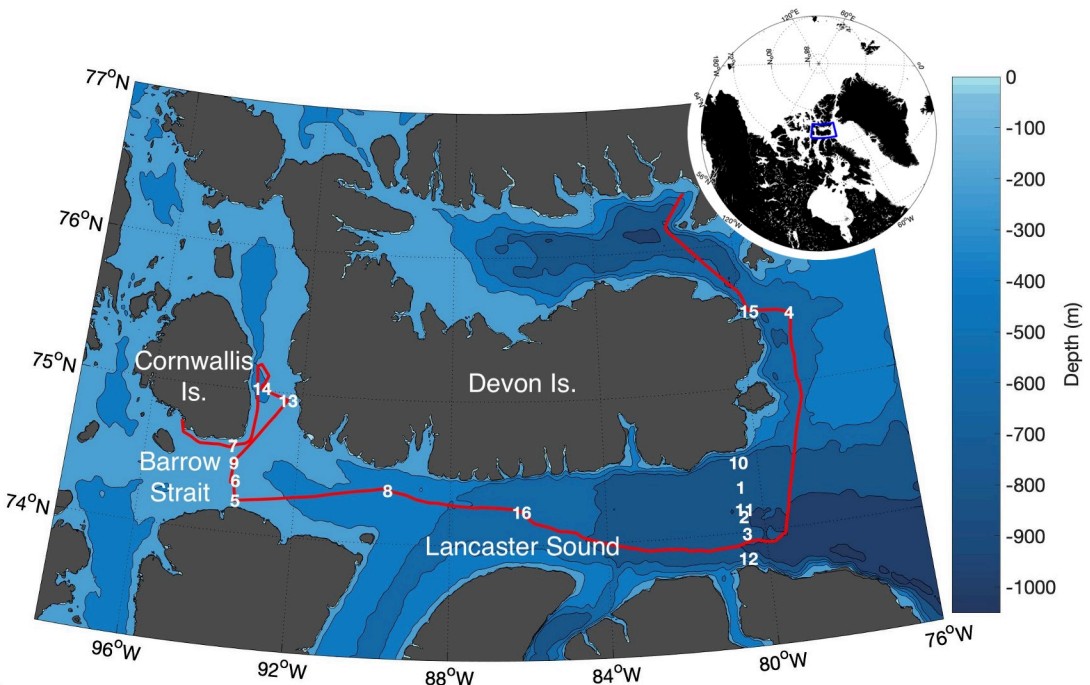

**Fig 1. Study region map.** The inset globe shows the geographic region of interest outlined in blue. The detailed map shows the numbered CTD stations sampled in August 2019, overlayed on bathymetric contours. We use the 200m bathymetric contour line around 92˚W to separate Lancaster Sound and Barrow Strait. The ship track is shown in red. The complete list of geographic coordinates sampled is available in the Polar Data Catalogue (doi: 10.5884/12715). The map was produced in Matlab (R2020a) using the publicly available m_map package [41].

derived estimates of the functional absorption cross section of PSII ($\sigma_{\text{PSII}}$, $\sigma_{\text{PSII}}$'), the minimum and maximum fluorescence when the reaction center pool is fully open and closed ($F_o$, $F$' and $F_m$, $F_m$', respectively), and the variable fluorescence ($F_v = [F_m - F_o]$; $F_q' = [F_m' - F']$). Single Turnover excitation flashlets were delivered simultaneously by 445, 470, 505, 535, and 590 nm

**Table 1. Commonly referred to photophysiological terms and abbreviations.**

| Term | Definition | Units |
|---|---|---|
| Chl$a$ | Chlorophyll $a$ fluorescence | Relative units |
| $F_o$, $F$' | Minimum Chl$a$ measured in dark and light acclimated state, respectively. | Relative units |
| $F_m$, $F_m$' | Maximum Chl$a$ measured in dark and light acclimated state, respectively. | Relative units |
| $\sigma_{\text{PSII}}$, $\sigma_{\text{PSII}}$' | Functional absorption cross-section for PSII in the dark and light acclimated state, respectively | Å$^2$PSII$^{-1}$ |
| $F_v/F_m$ | Photochemical yield in the dark acclimated state; $(F_m - Fo)/Fm$ | Dimensionless |
| $F_q'/F_{m^{150}}'$ | Photochemical yield measured under 150 µmol quanta m$^{-2}$ s$^{-1}$ | Dimensionless |
| $F_q'/F_{m^{Emax}}'$ | Photochemical yield measured under super saturating irradiance | Dimensionless |
| NPQ | Non-photochemical quenching; $(F_m - F_m')/F_m$ | Dimensionless |
| $\tau_{Qa}$ | Photosynthetic electron turnover rate measured at saturating irradiance | s$^{-1}$ |
| p | Probability of energy transfer between RCIIs | Unitless |
| $C_{Qa}(t)$ | Fraction of initially available RCIIs closed by excitation flashlets | Unitless |
| PAR | Photosynthetically active radiation | µmol quanta m$^{-2}$ s$^{-1}$ |
| $E(t,\lambda)$ | Transient PAR provided by excitation flashlets | µmol quanta m$^{-2}$ s$^{-1}$ |
| ETR$_a$ | Amplitude-based electron transport rate | e$^-$ s$^{-1}$ RCII$^{-1}$ |
| ETR$_k$ | Kinetic-based electron transport rate | e$^-$ s$^{-1}$ RCII$^{-1}$ |

LED lamps. Each curve fit and acquisition was based on fluorescence yields averaged from five sequential ST flashes, each with a total duration of 0.2 s. Each 5 flash sequence was separated by a 1s interval. Curve fitting and fluorescence averaging was completed within LIFT software (Soliense Inc.).

Each seawater sample was initially held in the cuvette under low light for five minutes to allow NPQ relaxation. During this low-light acclimation period, five LEDs at peak wavelength excitation of 445, 470, 505, 535, and 590 nm each supplied 1 μmol quanta $m^{-2}$ $s^{-1}$ of actinic light, providing a total irradiance of 5 μmol quanta $m^{-2}$ $s^{-1}$. Samples were acclimated to low light rather than total darkness to avoid RCII closure and fluorescence quenching associated with back flow of electrons from the PQ pool to $Q_a$ (e.g. [45]). After this short acclimation period, 40 acquisitions (each consisting of 5 averaged fluorescence transients) were collected. Irradiance was then increased to 150 μmol quanta $m^{-2}$ $s^{-1}$ (30 μmol quanta $m^{-2}$ $s^{-1}$ per LED) and samples were held for an additional five minutes to acclimate to the higher irradiance level before collecting another 40 acquisitions to measure photophysiology close to the saturation irradiance for primary photochemistry, previously found to range from 96–213 μmol quanta $m^{-2}$ $s^{-1}$ for surface assemblages in the Canadian Arctic during summer [46]. The LIFT-FRRf cuvette was then flushed for 60 s with seawater from the ship seawater supply, displacing at least five full cuvette volumes before the pump was turned off to isolate the next sample for measurements.

Semi-continuous sampling was interrupted every 90 mins to perform light response curves on static samples. For each light response curve, the actinic irradiance supplied by each LED lamp increased incrementally from 0 to 350 μmol quanta $m^{-2}$ $s^{-1}$ for all 5 LEDs to create light steps of 0, 15, 35, 75, 110, 150, 200, 300, 400, 550, 850, 1250, and 1750 μmol quanta $m^{-2}$ $s^{-1}$ total actinic light. A total of 25 acquisitions (of 5 averaged fluorescence transients each), were obtained at each light step with a pause of 30s at each new light level to provide some acclimation time. Fluorescence amplitudes generally stabilized by the third round of data acquisition, and we thus excluded the first two flash sequences at each light level from data analysis. Light response curves were used to calculate ETR$_{PSII}$ as a function of increasing light intensity (see below for calculation details). To produce final photosynthesis-irradiance curves, ETR$_{PSII}$ was plotted against irradiance and fit to the hyperbolic function described by Webb et al. [47], using a least squares non-linear regression to derive the maximum light-saturated photosynthetic rate (ETR$_{max}$), the saturating light intensity (E$_k$) and the light utilization efficiency ($\alpha$).

Daily blank corrections were performed by analyzing seawater gently passed through a 0.2 μm syringe filter (e.g. [25]). Soliense software was programmed to subtract the resulting fluorescence intensity from all measurements. Prior to the field deployment, the LIFT-FRRf lamps used to produce actinic light and the probing flashes were calibrated using a Walz spherical submersible micro quantum sensor (Walz, US-SQS-L).

## Electron Transport Rates (ETR$_{PSII}$)

The amplitude and kinetics-based ETR$_{PSII}$ algorithms (ETR$_a$ and ETR$_k$, both e$^-$ s$^{-1}$ RCII$^{-1}$) were applied to fluorescence data collected during light response curves to determine ETR$_{PSII}$ at increasing actinic light intensities. ETR$_a$ was applied to all 85 light response curves collected along the ship-track to produce photosynthesis-irradiance curves following Webb et al. [47]. At each light level, ETR$_a$ (e$^-$ RCII$^{-1}$ s$^{-1}$) was determined as the quantity of incident light absorbed by PSII directed towards photochemistry [27, 28];

$$\text{ETR}_a = PAR \times \sigma_{PSII} \times (F'_q/F'_m)/(F_v/F_m) \times 6.022 \times 10^{-3}, \tag{1}$$

Here, PAR ($\mu$mol quanta m$^{-2}$ s$^{-1}$) is the total actinic light provided by the FRRf LEDs, $\sigma_{PSII}$ (Å$^2$ PSII$^{-1}$) is the PSII functional absorption cross section measured in dark acclimated samples, and $F'_q/F'_m$ divided by $F_v/F_m$ (dimensionless) is the PSII photochemical efficiency measured under actinic light normalized by the dark-measured maximum photochemical efficiency. The constant 6.022$^*$10$^{-3}$ converts $\sigma_{PSII}$ units from Å$^2$ PSII$^{-1}$ to m$^2$ PSII$^{-1}$ and PAR from $\mu$mol quanta to quanta.

For ETR$_k$ calculations, fluorescence data from a subsample of 25 light response-curves with optimal signal-to-noise ratios were re-analyzed using a 3-component multi-exponential model to describe $Q_a$ reoxidation kinetics [23]. This numerical procedure was applied as a fitting option in the FRRf Soliense Software.

$$F(t) = F_o + (F_m - F_o)C_{Qa}(t)\frac{1-p}{1-C_{Qa}(t)p},\tag{2}$$

$$\frac{\partial C_{Qa}}{\partial t} = E(t,\lambda) \times \sigma_{PSII} \times \frac{1-C_{Qa}(t)}{1-C_{Qa}(t)p} - C_{Qa}(t)\sum_{i=1}^{3}\alpha_i/\tau_i,\tag{3}$$

Here, F(t) represents the measured fluorescence signal, $C_{Qa}(t)$ (dimensionless) is the fraction of available PSII reaction centers closed by excitation flashlets, with $C_{Qa}(t)$ equal to 1 when the photochemistry pathway is fully closed. The term p (dimensionless) is the connectivity factor, which describes the likelihood of energy transfer between RCIIs. $\alpha_i$ and $\tau_i$ refer to the amplitude and time constant of the ith component of $Q_a$ reoxidation, respectively. The value $\frac{\partial C_{Qa}}{\partial t}$ is determined by the balance between primary photochemistry induced by excitation flashlets (E; $\mu$mol quanta m$^{-2}$ s$^{-1}$) and electron transfer from $Q_a$ to secondary electron acceptor, $Q_b$, mediated by three kinetic components. Fluorescence transients were fit to retrieve photophysiological parameters by integrating Eq 3 over the length of the ST flash sequence and then iteratively fitting Eq 2. to the fluorescence data. Importantly, we note Eq 3 differs slightly from that applied by FIRe-based data analysis, in which there is an added term to describe RCII closure induced by actinic irradiances (See 'Computational Considerations' in the Discussion).

The rate of $Q_a$ reoxidation, $\tau_{Qa}$, was calculated by averaging the time constants of the two primary components of electron transfer from $Q_a$ to secondary electron acceptor $Q_b$ following the approach of Gorbunov and Falkowski [30]. The kinetic-based ETR$_{PSII}$ was then calculated as,

$$ETR_k = \frac{1}{\tau}\frac{PAR \times F'_q/F'_m}{PAR_{max} \times F'_q/F'_{mEmax}},\tag{4}$$

Here, PAR$_{max}$ is a super-saturating light level chosen as a value three–fold higher than the light saturation parameter, $E_k$, derived from photosynthesis irradiance curves. $F'_q/F'_{mEmax}$ is the PSII photochemical efficiency measured under PAR$_{max}$. The photosynthetic turnover rate, $\frac{1}{\tau}$ (s$^{-1}$), is taken as $\frac{1}{\tau_{Qa}}$ at saturating irradiance where primary photochemistry is at a maximum [30].

## Non-Photochemical Quenching (NPQ)

We quantified NPQ as a measure of the relative increase in heat dissipation of absorbed energy by PSII between samples exposed to low light and 150 $\mu$mol quanta m$^{-2}$ s$^{-1}$, following Bilger

and Bjorkman [48] as:

$$\mathrm{NPQ_{SV}} = (F_m - F'_m)/F_m, \tag{5}$$

This derivation assumes full relaxation of all NPQ processes in the dark-acclimated state. However, as discussed in the Results section, this assumption did not hold in our samples. For this reason, we used an additional measure of long-term NPQ processes, based on the ratio of photochemical efficiency measured under low and high light (i.e. $(F'_q/F'_m)/(F_v/F_m)$).

## Statistical analysis

Pair-wise statistical relationships between measured variables were determined using Spearman Rank correlation tests. Samples from two regions of our transect (Lancaster Sound and Barrow Strait) were compared using Kruskal-Wallis test. Lilliefors test rejected the null hypothesis that underway data were normally distributed, so we report median rather than mean values of all photophysiological variables. Deviation from the median was determined as the median absolute deviation. All analyses were completed using Matlab (Mathworks, R2020a).

# Results

## Hydrographic properties

Sea surface temperature (SST) and salinity varied significantly across our study region, reflecting the influence of different water masses and freshwater inputs. Sea surface temperature ranged from 0.6 to 15.5°C, and salinity ranged from 22.8 to 31.6 PSU. Salinity and SST strongly covaried, and exhibited a number of sharp transitions across prominent hydrographic fronts associated with freshwater input (Fig 2; S1 Table). Surface layer (~7m) phytoplankton biomass, approximated by in-situ Chl*a* fluorescence measurements (non-FRRf), was low throughout the entire transect, with a mean Chl*a* fluorescence equivalent to 0.15 ± 0.04 mg m$^{-3}$ (n = 7200). These in situ Chl*a* measurements showed significant diel periodicity, likely reflecting daytime fluorescence quenching effects. Both SST and salinity exhibited weak correlations with Chl*a* (S1 Table). Mixed layer depths recorded at the 16 profiling stations were shallow, with a mean of 9.2 ± 4.5 m. Mixed layer nitrate and nitrite concentrations at profiling stations were frequently below the detection limit (S2 Table, mean 0.03 ± 0.049 μM, n = 16), indicative of post-bloom, nutrient-limited summer conditions. The mixed layer Chl*a* concentration averaged across all 16 profiling stations was 0.40 ± 0.18 mg m$^{-3}$ (n = 16). There was no statistically significant relationship between station Chl*a* and mixed layer nitrate and nitrite concentrations ($\rho$ = 0.09, p = 0.75, n = 16). Biological oxygen saturation, derived from ΔO2/Ar measurements, was greater than zero across the entire cruise track, implying net autotrophic conditions. Small-scale features in ΔO2/Ar distributions were observed across hydrographic frontal regions with only weak correlation to salinity or temperature measured along the cruise track (S1 Table).

## Photophysiological measurements

**Continuous flow-through measurements.** The photophysiological parameters $F_v/F_m$ and $\sigma_{PSII}$ fluctuated over diel cycles throughout our sampling period (Fig 3a and 3b), showing daily maxima during the night, and decreasing during daylight hours. Both of these variables exhibited a significant negative correlation with actinic surface PAR intensity averaged over the 5 min window prior to sample measurements (Table 2; Fig 4a and 4b). This result is consistent with previously noted in-situ photophysiological diurnal patterns of daytime fluorescence

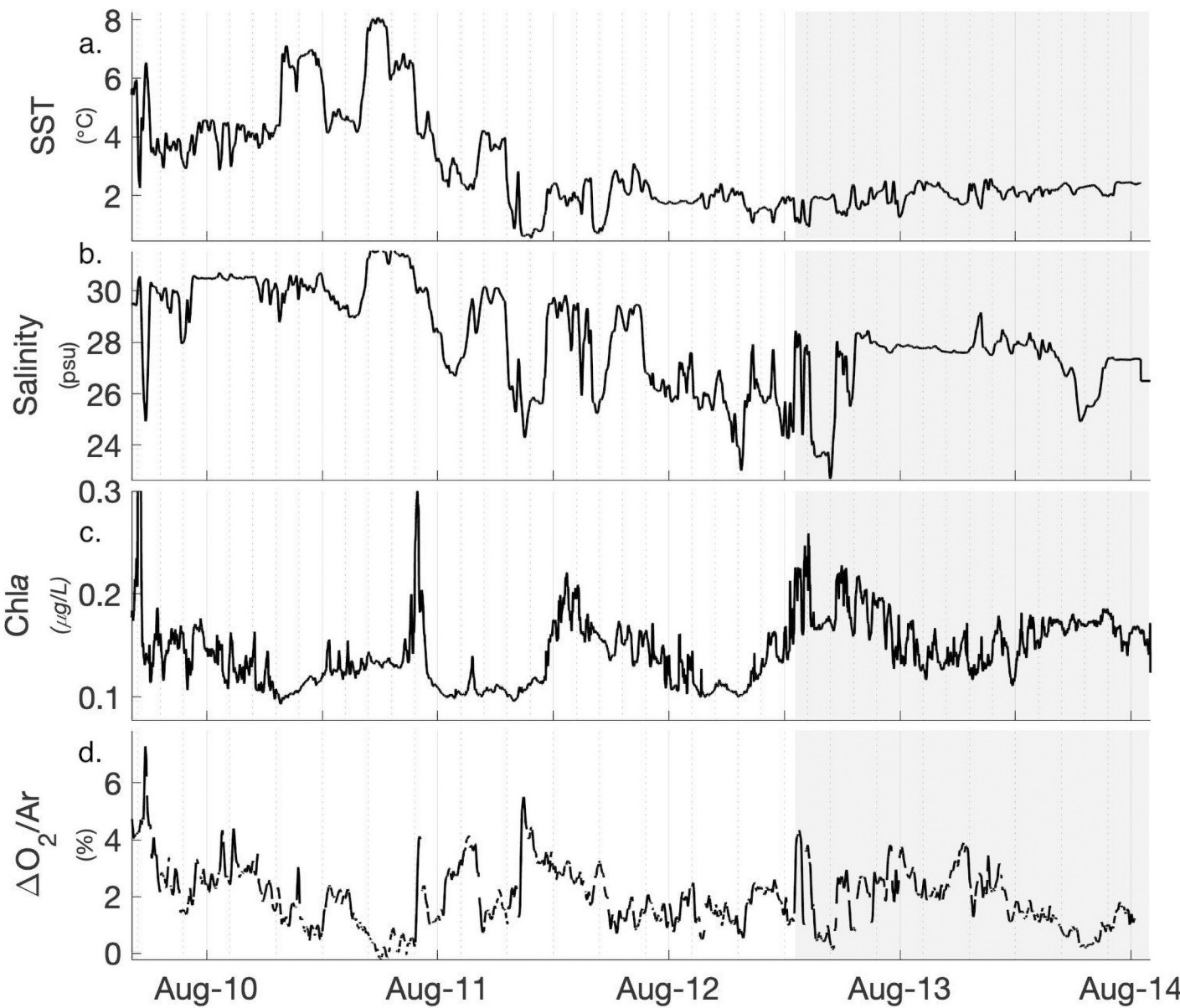

**Fig 2. Summary of oceanographic conditions.** Hydrographic conditions, phytoplankton biomass and biological oxygen saturation measured in surface waters (∼ 7 m depth) along the cruise track. Shaded area on Aug. 13–14 marks measurements made in Barrow Strait.

quenching [49, 50]. We note, however, that these $F_v/F_m$ and $\sigma_{PSII}$ measurements were collected after 5 minutes of low light exposure, indicating the persistence of longer-lived light-dependent quenching effects after this acclimation period. We thus conclude that dark-acclimation (i.e. NPQ relaxation) was not achieved in our measurement protocol, and that $F_v/F_m$ and $\sigma_{PSII}$ are thus representative of an intermediate state between in-situ and dark-acclimated values.

Despite signs of strong light-dependent quenching, NPQ values, calculated as $NPQ_{SV}$ = (Fm − Fm')/Fm, were negatively correlated to surface PAR (Fig 4c). This surprising result can be explained by the derivation used here for NPQ, which measures the fractional change in NPQ between light and dark (low light) measurements, rather than total NPQ. As a result, this

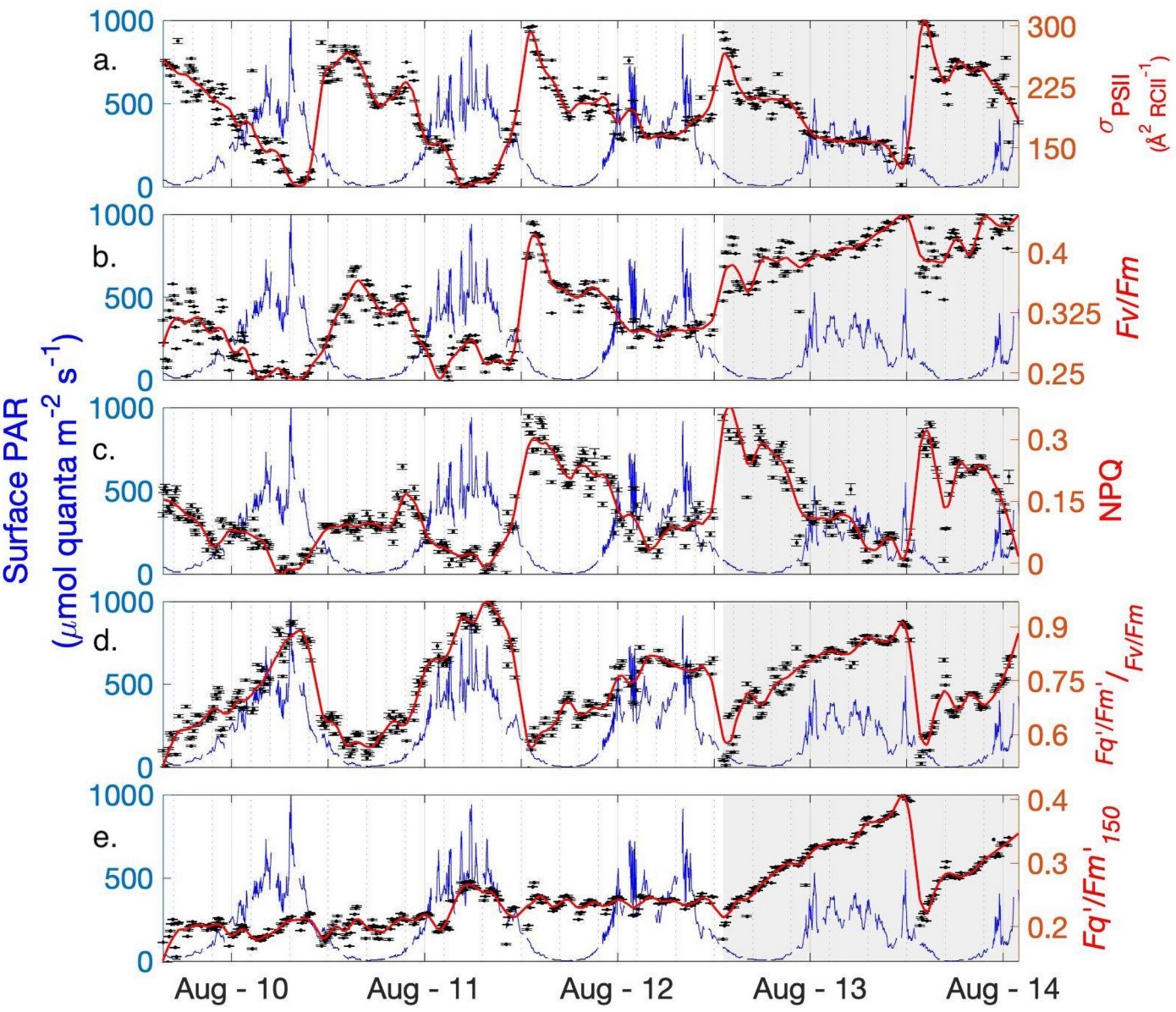

**Fig 3. Summary of photophysiological conditions.** Semi-continuous FRRf measurements of photophysiology (black dots) are superimposed over the in-situ surface PAR (blue line). A loess smoothing function was applied to photo-physiological measurements (red line). Shading denotes the Barrow Strait portion of the transect.

approach does not account for any residual quenching present in samples after five minutes of low light acclimation. To address this limitation, we used the ratio $(F'_q/F'_{m^{150}})/(F_v/F_m)$ to estimate the extent of residual quenching present in samples after five minutes of NPQ relaxation. As expected, this derived variable was well correlated to surface PAR (Fig 4d, Table 2), reflecting the effects of residual quenching and, potentially, short-term photoinhibition.

To further examine low light-acclimated $F_v/F_m$ and $\sigma_{PSII}$ values, we isolated night-time $F_v/F_m$ and $\sigma_{PSII}$ measurements collected under relatively low ambient surface PAR. Due to the long summer daylight hours in the Arctic, only 1% of data points represents true night when surface PAR = 0. We thus chose light levels $\leq$ 100 μmol quanta m$^{-2}$ s$^{-1}$ to represent night-time

**Table 2. Correlation of underway photophysiological variables and surface PAR.**

| Photophysiological variables | Surface PAR | | |
|---|---|---|---|
| | **Lancaster Sound**<br>**n = 283** | **Barrow Strait**<br>**n = 198** | **Total**<br>**n = 481** |
| $F_v/F_m$ | $\rho = -0.72^{**}$ | $\rho = 0.27^{**}$ | $\rho = -0.72^{**}$ |
| $\sigma_{PSII}$ | $\rho = -0.79^{**}$ | $\rho = -0.60^{**}$ | $\rho = -0.50^{**}$ |
| NPQ | $\rho = -0.66^{**}$ | $\rho = -0.65^{**}$ | $\rho = -0.65^{**}$ |
| $F'_q/F'_{m150} : F_v/F_m$ | $\rho = 0.81^{**}$ | $\rho = 0.66^{**}$ | $\rho = 0.68^{**}$ |
| $F'_q/F'_{m150}$ | $\rho = 0.21^{**}$ | $\rho = 0.67^{**}$ | $\rho = -0.01$ |

Spearman rank correlations between underway photophysiological measurements and recent surface PAR exposure, analyzed by region and for the total dataset.

$^{**}$ is used to indicate p values $\leq 0.01$.

conditions. By comparison, midday surface PAR ranged from 400–1000 μmol quanta m$^{-2}$ s$^{-1}$. The median night-time $F_v/F_m$ for the entire transect was 0.36 ± 0.03 (n = 214), a value similar to the global average of 0.35 ± 0.11 [51], but lower than the 0.55 median value previously recorded for late-summer assemblages in the Canadian Arctic [52]. The lowest $F_v/F_m$ values were recorded at the beginning of the ship track (August 10–12) within Lancaster Sound

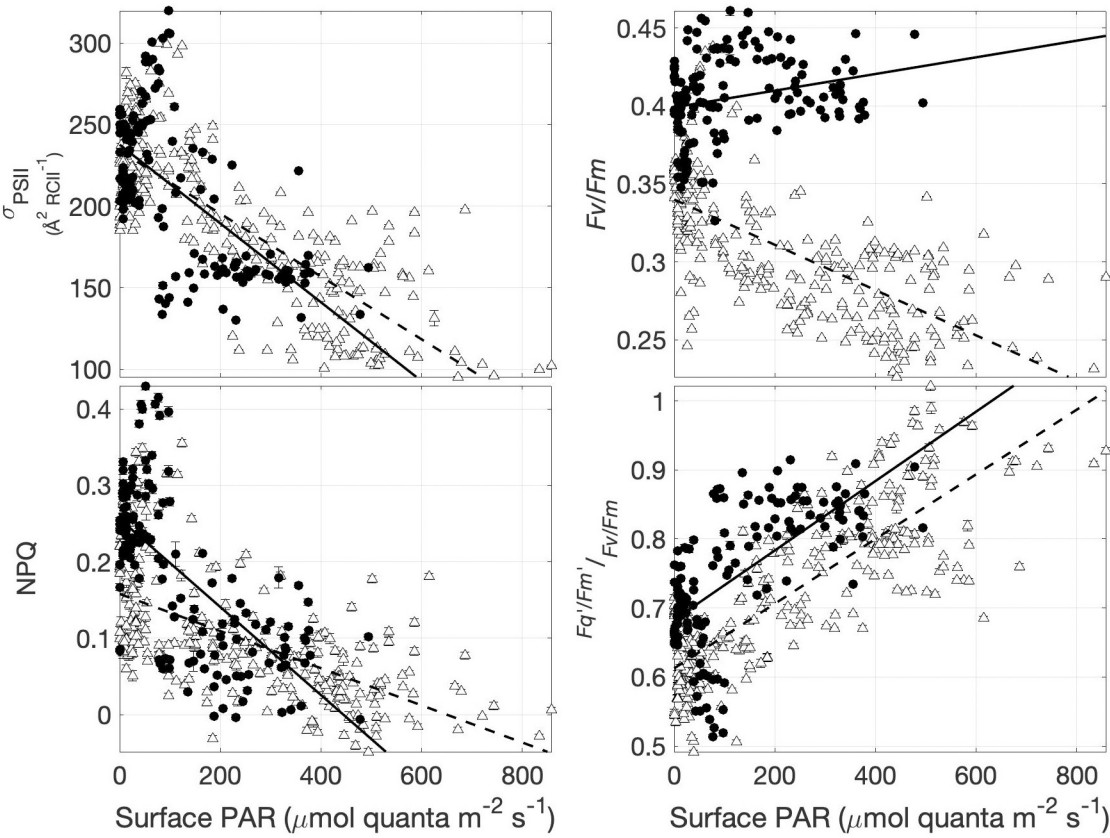

**Fig 4. Recent light history effects on photophysiology measured under low light.** FRRf-derived photophysiological parameters measured under low light are plotted against in-situ surface PAR at the time of sample acquisition. Panels (a) and (b) show $F_v/F_m$ and $\sigma_{PSII}$ measurements made after 5 min of low light exposure. Panels (c) and (d) show NPQ and residual quenching measured after low light treatment. Lines of best fit are shown for Barrow Strait (solid line) and Lancaster Sound (dashed line). Full regression analyses results are reported in S2 Table. Error bars show standard error, but are often concealed by size of data symbols.

$(0.32 \pm 0.03$, n = 120), while $F_v/F_m$ increased significantly in Barrow Strait $(0.40 \pm 0.02$, n = 94), indicating greater photosynthetic potential in this region (Fig 3b). In contrast, night-time $\sigma_{PSII}$ did not significantly vary between Lancaster Sound $(250 \pm 17.2)$ and Barrow Strait $(241.1 \pm 15.3)$ (Fig 3a). Note that these absolute $\sigma_{PSII}$ values are somewhat lower than those reported in previous studies, likely reflecting our use of simultaneous excitation flashlets centered around 445, 470, 505, 535, and 590 nm. Relative to blue light, not all of these wavelengths are efficiently absorbed by phytoplankton, resulting in an apparent decrease $\sigma_{PSII}$ [53]. As discussed below, $\sigma_{PSII}$ values are also subject to physiological, taxonomic and environmental effects [54].

As observed for $F_v/F_m$, the photosynthetic efficiency measured under 150 µmol quanta m$^{-2}$ s$^{-1}$ actinic light $(F_q'/F_{m'150})$ displayed strong regional differences, increasing from $0.21 \pm 0.01$ (n = 283) in Lancaster Sound to $0.30 \pm 0.02$ (n = 198) in Barrow Strait. However, unlike the low-light measurements, $F_q'/F_{m'150}$ did not exhibit a diel signature and was remarkably consistent across samples measured within Lancaster Sound, where $F_q'/F_{m'150}$ was weakly correlated to surface PAR (Table 2; Figs 3e and 5). Whereas, $F_q'/F_{m'150}$ in Barrow Strait was strongly positively correlated to surface PAR (Table 2). Such regional differences in the $F_q'/F_{m'150}$, surface PAR relationship imply differential capacities of Lancaster Sound and Barrow Strait phytoplankton to photoacclimate to higher irradiances.

**Photosynthesis-irradiance curves and ETR$_{PSII}$ comparisons.** We used light response curves to compare FRRf-based ETR$_a$ and ETR$_k$ estimates. In this approach, ETR$_a$ (Eq 1) was plotted against actinic irradiance (Fig 6) to derive the maximum rate of charge separation at RCII (ETR$_{max}$), the light-dependent increase in the charge separation rates ($\alpha$), and the saturating light intensity (E$_k$). Fit parameters from these curves varied considerably, with mean

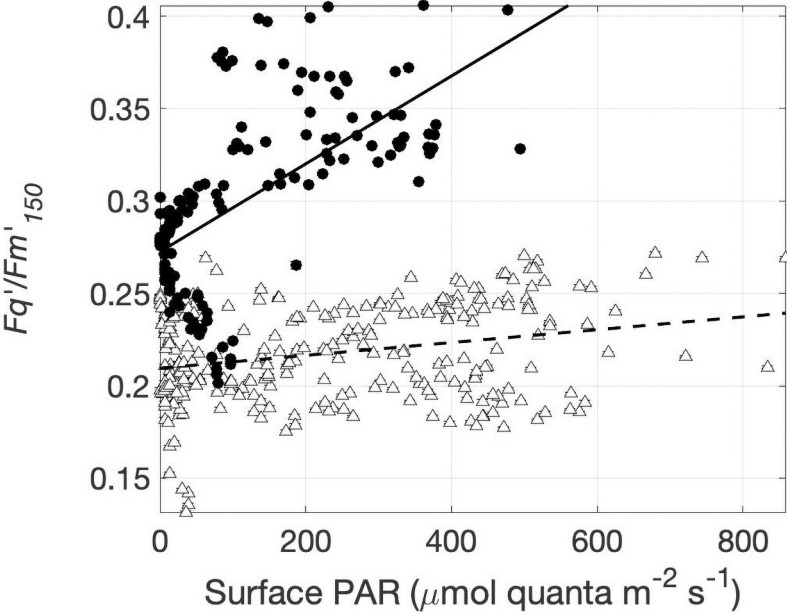

**Fig 5. Recent light history effects on photophysiology measured under low light.** The PSII photochemical efficiency measured under 150 µmol quanta m$^{-2}$ s$^{-1}$ in relation to natural surface irradiance at the time of sample acquisition. Lines of best fit are shown for Barrow Strait (solid line) and Lancaster Sound (dashed line). Full regression analyses results are reported in S2 Table. Error bars show standard error, but are often concealed by size of data symbols.

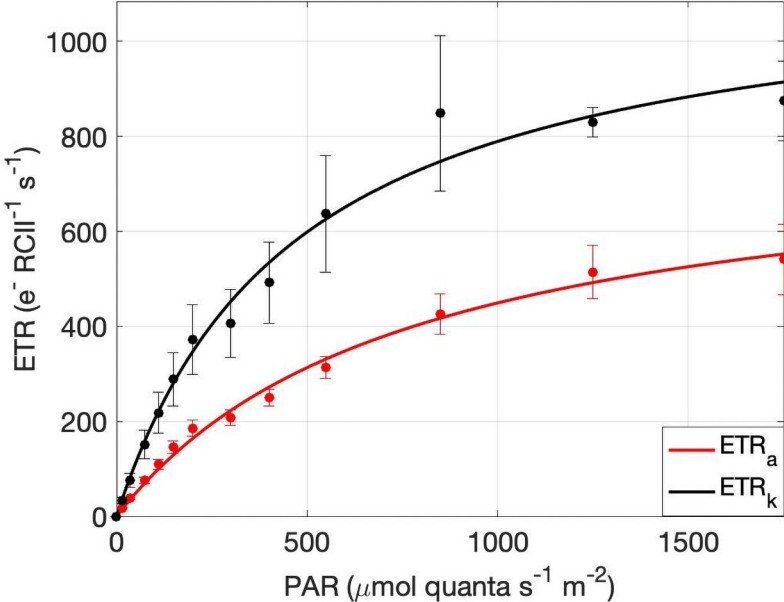

**Fig 6. Photosynthesis-irradiance curves derived by ETR$_a$ and ETR$_k$.** Consolidated mean ETR$_a$ (red) and ETR$_k$ (black) estimates at each light step of the 25 reprocessed P-I curves. Error bars represent the standard error of all individual measurements from all curves at each light step. Curves were produced using the photosynthesis-irradiance function described by Webb et al. (1974).

values of ETR$_{max}$, $\alpha$, and E$_k$ of 460 ± 345 e$^-$ s$^{-1}$ RCII$^{-1}$, 1.37 ± 0.57 e$^-$ RCII$^{-1}$ quanta$^{-1}$ m$^{-2}$, and 358.0 ± 199.0 μmol quanta m$^{-2}$ s$^{-1}$, respectively (n = 85).

A subsample of 25 light response curves was re-analyzed using the 'kinetic' approach (ETR$_k$; Eq 5), and compared with ETR$_a$ values. This comparison revealed a strong correlation between the ETR$_{PSII}$ values ($\rho$ = 0.81, p < 0.001; Fig 7). However, the kinetics-based algorithm produced consistently higher results than ETR$_a$, with values 1.96 ± 1.2 times greater, on average, than ETR$_a$ (Fig 6).

Along our cruise track, we observed a distinct spatial pattern in the ETR$_k$:ETR$_a$ ratio. The highest values (3.26 ± 0.95, n = 14) occurred during the early part of our survey (August 10–13), with significantly lower ETR$_k$:ETR$_a$ (1.42 ± 0.16, n = 11) observed near the end of our cruise, particularly in Barrow Strait where $F_v/F_m$ was at its maximum (Fig 8). More generally, ETR$_k$:ETR$_a$ displayed a strong negative correlation with $F_v/F_m$ values ($\rho$ = -0.60, p < 0.01, n = 25). As discussed below, this result suggests ETR$_k$:ETR$_a$ decoupling is strongly driven by photophysiological and environmental variability.

## Discussion

The primary focus of our work was to quantify phytoplankton photophysiology and photo-chemical yields along our ship-track using active Chl*a* fluorescence methods. With this in mind, we applied a sampling and analysis strategy to support both amplitude-based and kinetic analysis of FRRf data to derive ETR estimates. In the following, we first discuss the spatial patterns in FRRf-based measurements in relation to nutrient concentrations and light histories of phytoplankton assemblages encountered along our sampling transect. We then examine potential factors leading to the uncoupling of ETR$_{PSII}$ estimates, including environmental and physiological variability, and potential influences of different data analysis

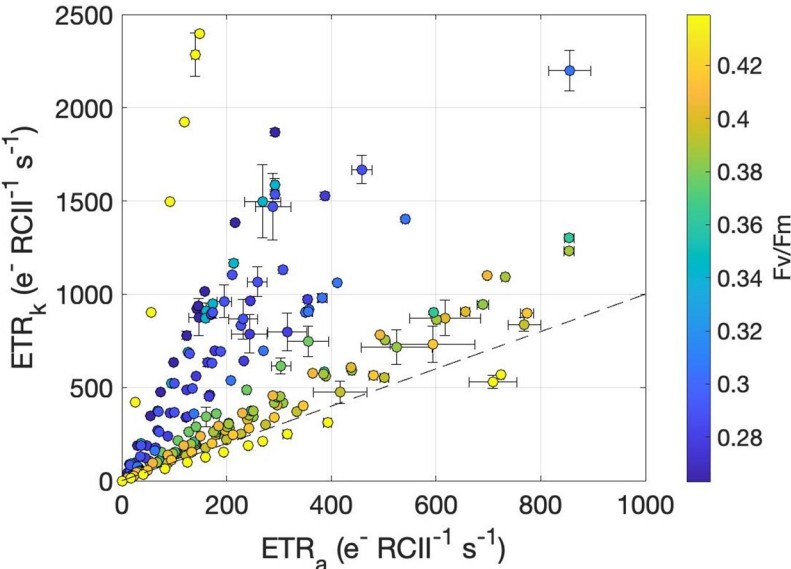

**Fig 7. $ETR_k$ plotted against $ETR_a$ derived from photosynthesis-irradiance curves and colored by $F_v/F_m$.** Each point represents the mean ETR value at a given light intensity within a Photosynthesis-Irradiance curve, and error bars are the standard error. The dashed line indicates a 1:1 relationship.

methods. We conclude by discussing the implications of our results for future ship-board FRRf deployments.

## Spatial variation of photophysiology

Across our survey region, we observed notable spatial patterns in FRRf data, with low $F_v/F_m$ and $F'_q/F'_{m150}$ values in Lancaster Sound, and significantly higher values in Barrow Strait. Low Fv/Fm values are typical in the late-summer Arctic, and have been shown to increase in response to nitrate (but not phosphate) enrichments [5, 55]. This result, coupled with the

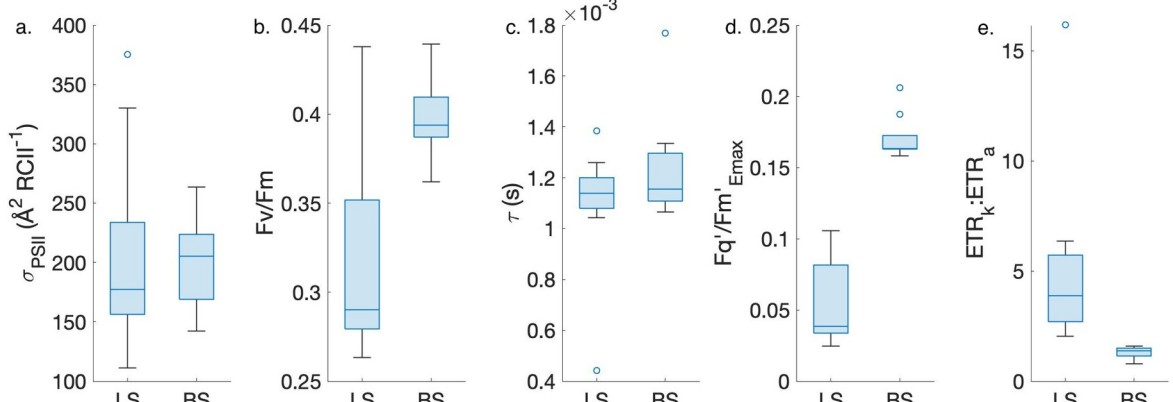

**Fig 8. Parameters contributing to the $ETR_k$:$ETR_a$ ratio measured in Lancaster Sound (LS) versus Barrow Strait (BS).** Horizontal lines within each boxplot represent the median. The upper and lower edge of each box demarks upper and lower quartiles, respectively, while whiskers extend over the entire data range, excluding outliers. Outliers, determined as data points falling over 1.5x the interquartile distance away from box edges, appear as unfilled circles. p values in each subplot are results from 2-group Kruskal-Wallis tests. All data shown here was collected during the 25 reprocessed P-I curves.

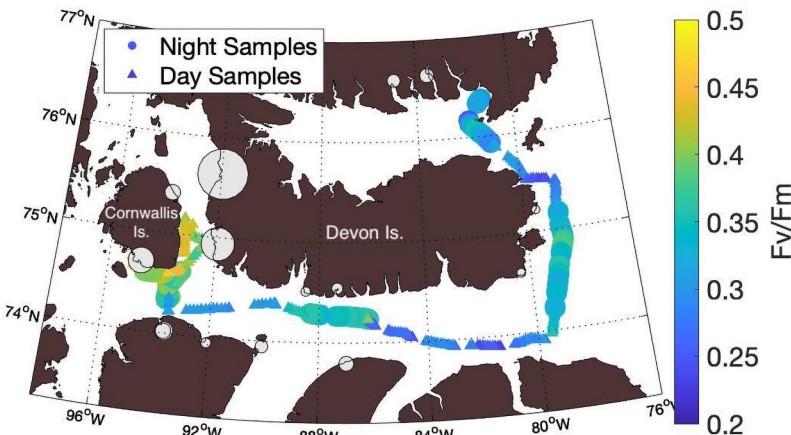

**Fig 9. Spatial distribution of riverine nutrient inputs and photochemical efficiency ($F_v/F_m$) along the ship track.** River contributions of nitrate and nitrite are indicated by the size of grey bubbles. The largest inputs of nitrate and nitrite in the region are concentrated in the strait between Cornwallis and Devon Islands, coincident with observations of raised $F_v/F_m$ values. Larger circles are used to denote night-time measurements of $F_v/F_m$, whereas smaller triangles denote day-time measurements.

nutrient depletion observed at profiling stations along our cruise track (Table 2), suggests that the low $F_v/F_m$ values we observed likely reflect nitrogen deficiency. Iron (Fe)-limitation has also been shown to exert a strong negative effect on $F_v/F_m$, coincident with increases in $\sigma_{PSII}$ values. These Fe-dependent effects result from a combination of physiological responses [36, 56, 57] and taxonomic shifts towards smaller cells [54]. In contrast, the low $F_v/F_m$ values recorded in the Lancaster Sound were not associated with high $\sigma_{PSII}$, and correlation analysis of night-time $\sigma_{PSII}$ and $F_v/F_m$ values revealed a weak positive relationship between $\sigma_{PSII}$ and $F_v/F_m$ ($\rho = 0.17$, p < 0.05, n = 214). Based on these results, and the proximity of our sampling to land-based Fe sources [58], we ruled out iron limitation as a likely cause for the low photo-efficiencies observed from August 10–13 in Lancaster Sound.

Notwithstanding the low background nutrient concentrations measured at profiling stations, chemical analyses from the Canadian Arctic Archipelago Rivers Program and Canadian Arctic GEOTRACES program have revealed elevated nitrate and nitrite concentrations within several rivers that discharge into Barrow Strait (Fig 9; [11, 59]). In this region, we observed low surface water salinity and elevated $F_v/F_m$, suggesting a link between river input and increased photosynthetic efficiency, which we ascribe to nutrient inputs. Additionally, the greatest mixed layer depths were found at the two CTD profiling stations situated between Cornwallis Island and Devon Island (Table 2), indicative of enhanced mixing associated with strong tidal currents, in agreement with model predictions of elevated mixing in Barrow Strait [10]. These observations suggest that spatial differences observed in FRRf-derived photophysiology may reflect elevated nutrient availability in Barrow Sound, resulting from a combination of river inputs and mixing effects.

Several mechanisms can explain the apparent effects of increased nitrogen availability on phytoplankton photophysiology. First, increased nitrogen availability enables protein synthesis needed to repair inactive reaction centers [60, 61]. Indeed, phytoplankton in Barrow Strait appeared to have an increased ability to acclimate to higher light levels, as evidenced by the strong positive relationship between $F'_q/F'_{m^{150}}$ and surface PAR ($\rho = 0.67$, p < 0.01, n = 94; Fig 5) observed in this region. By comparison, $F'_q/F'_{m^{150}}$ exhibited only a weak relationship to surface PAR in Lancaster Sound where $F'_q/F'_{m^{150}}$ was significantly reduced and relatively constant

(Fig 5). This result suggests that, phytoplankton in Lancaster Sound had a reduced capacity to maintain protein repair rates needed to prevent lasting photodamage at higher light levels.

In addition to direct physiological effects, localized nutrient loading may indirectly affect FRRf signatures by stimulating a shift from small to larger phytoplankton species, for instance from nano-flagellates to diatoms [55]. Unfortunately, we lack information on phytoplankton assemblage composition, and thus cannot directly examine any potential taxonomic effects on photophysiological signatures. However, such a shift from small to large cells would be expected to drive a decrease in $\sigma_{PSII}$, concurrent with increases in $F_v/F_m$ [54], which we did not observe. Previous pigment analyses conducted in the Canadian Arctic Archipelago found that Lancaster Sound and Barrow Strait were both dominated by diatom species, followed by dino-flagellates, in summer [56]. We thus infer that the spatially-divergent $F_v/F_m$ values, coupled with persistently low $\sigma_{PSII}$, primarily reflect photophysiological nutrient effects in relatively large cells.

## Light-dependent effects and residual NPQ

We observed strong residual NPQ effects after five minutes of low light acclimation (Figs 3 and 4). Notably, the extent of these quenching effects was a predictable function of the short-term light history experienced by in situ phytoplankton assemblages (Fig 3d). Previous studies examining the drivers of NPQ variability [62–64] suggest that the magnitude of NPQ effects at a given light level is tied to a number of environmental factors (e.g. temperature and CO2 concentrations), phytoplankton taxonomy and physiological status. Given these sources of variability, NPQ relaxation times needed for robust $F_v/F_m$ measurements are expected to differ significantly across ocean regimes. In cold waters, such as those encountered along our ship track, NPQ relaxation is slower [65], and this may have contributed to the longer-lived quenching observed in our low-light samples. As the spatial and temporal resolutions of $F_v/F_m$ and ETR$_a$ measurements are constrained by such acclimation periods, it is recommended that future FRRf field deployments conduct experiments using natural assemblages to determine the regional minimum relaxation period necessary to achieve steady-state dark-acclimation. This acclimation step can then be incorporated into underway FRRf protocols, resulting in more robust ETR$_a$ estimates, albeit with reduced measurement frequency. Such routine determinations of the minimum NPQ relaxation time requirements have not been commonly carried out for marine phytoplankton [25]. As a result, there is little systematic knowledge of the global variability of NPQ relaxation times. Adopting such pre-study tests (or within protocol) as standard practice would improve current understanding of environmental and taxonomic controls on NPQ relaxation kinetics. Moreover, it may be necessary to adjust the length of the dark-acclimation period to reflect changing conditions over the duration of a cruise. We thus recommend that future work incorporate semi-regular assessments of dark-acclimation times into field-sampling protocols.

As an initial step towards resolving the potential influence of dark acclimation times on retrieved photo-physiological parameters, we conducted preliminary experiments on natural Arctic phytoplankton assemblages to determine optimal dark acclimation times, whereby $\sigma_{PSII}$ and $F_v/F_m$ recovery rates were determined for surface water samples retained in darkness over 30 minutes. These preliminary experiments were carried out on the most recent *CCGS Amundsen* expedition (Sept. 03–09, 2021) as the ship transited through Lancaster Sound.

Our analysis revealed that phytoplankton reached maximum, stable, levels of light utilization efficiency (as judged by $\sigma_{PSII}$ and $F_v/F_m$) within 9.65 ± 1.53 min (n = 34). In our present study, we used a constant 5 minute low light acclimation period. Increasing this to 10 minutes would have increased our total measurement interval from 12 to 17 minutes, resulting in

a $\sim$ 30% reduction in sample size. Ultimately, the appropriate low light/dark acclimation time depends on the research question of interest as sampling protocols with shorter acclimation periods provide higher resolution datasets more reflective of in-situ photophysiological conditions, whereas longer acclimation periods reduce NPQ effects on $\sigma_{PSII}$ and $F_v/F_m$ values, and thus simplify data interpretation.

## Decoupling of $ETR_a$ and $ETR_k$

Across our study region, $ETR_k$ significantly exceeded $ETR_a$ (Fig 6), with the magnitude of $ETR_k$ and $ETR_a$ decoupling varying strongly in response to phytoplankton photophysiological conditions (Fig 7). The ratio between $ETR_k$ and $ETR_a$ depends on a number of variables:

$$ETRk : ETR_a = \frac{F_v/F_m}{\sigma_{PSII} \times \tau \times F_q'/F_{mEmax}' \times Emax \times 6.022 \times 10^{-3}}, \tag{6}$$

Importantly, all of the terms defining $ETR_k$:$ETR_a$ are potentially responsive to shifts in nutrient abundances and phytoplankton taxonomic composition. Nutrient enrichment experiments have demonstrated increasing $\tau_{Qa}$ with nutrient deficiency and elevated actinic irradiance [30, 55]. Moreover, $\sigma_{PSII}$ can also change with nutrient availability, but the observed percent change in $\sigma_{PSII}$ following short-term nitrate enrichment is small compared with that of $F_v/F_m$ [5, 55]. Baseline fluorescence, may also influence the terms used to define $ETR_k$:$ETR_a$. This phenomenon represents a non-variable contribution to the Chl fluorescence signal, which is understood to reflect the presence of energetically decoupled light harvesting complexes under nutrient limitation or photoinhibitory stress [31]. High baseline fluorescence decreases the amplitude of fluorescence transients, making $F_v/F_m$ a useful gauge of phytoplankton physiological stress [5, 66, 67]. Analysis of our data revealed a strong correlation between $F_v/F_m$ and $F_q'/F_{mEmax}'$, ($\rho = 0.93$, p < 0.001, n = 25), suggesting $F_q'/F_{mEmax}'$ is similarly affected by baseline fluorescence and reflective of physiological status. However, in one sample, we recorded the highest $F_v/F_m$ but the lowest $F_q'/F_{mEmax}'$. The sample also showed the highest $ETR_k$:$ETR_a$ value (Fig 7). After removing this one sample, the correlation between $F_v/F_m$ and $ETR_k$:$ETR_a$ strengthened from $\rho = $ -0.60 before removal to $\rho = $ -0.86 after removal (p < 0.01 for both, n = 25 and 24, respectively). We conclude that differential environmental and taxonomic sensitivities of the variables used to derive $ETR_a$ and $ETR_k$ can lead to discrepancies between these two productivity photosynthesis metrics.

Further investigation of $ETR_k$ and $ETR_a$ divergence is critical to inform our understanding of electron requirements for carbon assimilation and biomass production, particularly under nutrient-limiting conditions [28]. For instance, Schuback et al. [36] found that iron-limited phytoplankton assemblages exhibited elevated $ETR_a$ and greater decoupling between $ETR_a$ and C-assimilation rates as compared to iron-enriched assemblages. This result was attributed to the higher $\sigma_{PSII}$ values in iron limited samples. Since $ETR_k$ does not directly include $\sigma_{PSII}$, we speculate that C-assimilation will show less decoupling with this photosynthesis metric under low iron conditions. This hypothesis remains to be tested in future studies.

## Computational considerations

Beyond the physiological and taxonomic effects described above, the computational procedures used to analyze Chl$a$ fluorescence relaxation kinetics and derive turnover rates of electron transport molecules may also have a direct effect on the observed relationship between $ETR_k$ and $ETR_a$. We derived $\tau_{Qa}$ parameters used to calculate $ETR_k$ using the FRRf fluorescence transient fitting approach, as outlined in 'Electron Transport Rates, $ETR_{PSII}$' Materials

and Methods section. This approach relies on numerically fitting the rate of change in the redox state of Qa, driven by electron fluxes in and out of PSII reaction centers.

$$\frac{\partial C_{Q_a}}{\partial t} = e_{in} - e_{out},$$

(7)

Here $e_{in}$ is equivalent to the rate of primary photochemistry induced by excitation flashlets, and $e_{out}$ is controlled by $Q_a$ reoxidation. Within FRRf Soliense software, $e_{in}$ is formulated as,

$$e_{in} = E(t, \lambda) \times \sigma_{PSII} \times \left(\frac{1 - C(t)}{1 - C(t)p}\right),$$

(8)

By comparison, FIRe-based analysis of fluorescence transients deviates from FRRf by including an additional term to describe reaction center closure by background actinic light (PAR) [30]. As a result, Eq 9 is modified as:

$$e_{in} = E(t, \lambda) \times \sigma_{PSII} \times \left(\frac{1 - C(t)}{1 - C(t)p}\right) + PAR(\lambda) \times \sigma_{PSII} \times \left(\frac{1 - C(t)}{1 - C(t)p}\right),$$

(9)

The FRRf based analysis does not include a PAR term, as it is presumed that constant background light influences the baseline of the fluorescence signal, but does not contribute to dynamic changes in fluorescence measured over the course of an ST flash [Z. Kolber, *pers. comm.*]. In this interpretation, C(t) represents the fraction of initially available reaction centers closed by excitation pulses, such that C(t = 0) always equals 0.

Gorbunov and Falkowski [30] conducted a primary analysis of differences in $\tau_{QA}$ values retrieved from FRRf and FIRe fluorescence relaxation analyses. Their results showed that when the effect of background light was explicitly included in numerical formulations, FIRe-derived $\tau_{QA}$ values displayed strong actinic light dependencies, increasing with irradiance until plateauing around saturating irradiances ($E_k$). By contrast, their FRRf-derived $\tau_{QA}$ values varied little with actinic irradiance and resulted in a markedly shorter photosynthetic turnover time. Our own FRRf-derived $\tau_{QA}$ values displayed a weak relationship with applied actinic irradiances ($\rho = 0.17$, p < 0.01, n = 203). It is possible that applying the FIRe model fit to our own data may have also yielded slower photosynthetic turnover times, and therefore lower $ETR_k$ estimates, but it is unclear to what extent the alternative model may have affected our $ETR_k$ results and the observed decoupling between $ETR_k$ and $ETR_a$.

Going forward, it will be important to separate physiological drivers of $ETR_k$ and $ETR_a$ decoupling from offsets resulting from the use of different mathematical numerical approaches to data analysis. Discrepancies between $ETR_k$ and $ETR_a$ that cannot be explained by different derivations of $\tau_{Qa}$ must be attributable to differences between the two $ETR_{PSII}$ algorithms themselves. This raises the important question of which approach is most accurate, as neither has been established as a "gold standard". Addressing this issue will require parallel independent measurements of PSII activity, such as gross oxygen evolution measurements from $^{18}O$ experiments, as preformed previously for $ETR_a$ (e.g. [54]) but not $ETR_k$. Such fluorescence-independent validations of $ETR_a$ and $ETR_k$ are critically lacking, and will help elucidate the taxonomic and environmental influences on FRRf-based productivity photochemistry measurements [28]. This, in turn, will be of significant practical utility to FRRf users seeking to derive ship-based primary productivity estimates.

## Conclusions and future recommendations

Fast Repetition Rate fluorometry offers a means to rapidly assess both physiological status and photosynthetic electron transport rates of phytoplankton. The aim of this study was to evaluate

an autonomous protocol for high resolution FRRf measurements of phytoplankton physiology, and to compare two alternative models for deriving primary productivity photochemistry estimates from FRRf data. Our results demonstrate significant residual NPQ effects after five minutes of low light acclimation, suggesting the need for extended low light acclimation periods, which would significantly decrease measurement frequency. Our findings also illustrated localized regions of elevated $F_v/F_m$, likely linked to local nitrogen loading by freshwater inputs and tidal mixing. Although the amplitude-based and kinetic-based derivations of photosynthetic electron transport rates were well correlated, absolute agreement between estimates appeared to be affected by phytoplankton photophysiology, with the two models diverging under nutrient-limited conditions.

As a first step, resolving discrepancies between $ETR_a$ and $ETR_k$ will require consensus regarding the analysis of raw fluorescence transient data. This aim is fundamental for consistent data reporting among the growing community of FRRf and FIRe users [31]. Second, $ETR_a$ and $ETR_k$ should be validated against fluorescence-independent measures of productivity. Considering that measurement of $ETR_k$ does not require a dark acclimation step, short ($\sim 5$ min.) acclimation steps should be sufficient to accurately derive this term. Based on these findings, the $ETR_k$ model, if validated against independent productivity metrics, may be advantageous for high resolution evaluations of in-situ photosynthetic rates under ambient light conditions.

## Supporting information

**S1 Table. Correlation of underway hydrographic variables.** Results of Spearman Rank correlation analyses between each underway hydrographic variable are displayed. ** is used to indicate p values < 0.001. In all instances n = 7200.
(PDF)

**S2 Table. CTD profiling stations.** Sampling station locations are indicated as LS for Lancaster Sound or BS for Barrow Strait. Station mixed layer depth (MLD), mean mixed layer nitrate and nitrite concentration, and Chl a concentration within the mixed layer are shown.
(PDF)

**S3 Table. Surface PAR and underway photophysiological variable regression analyses.** All p values were < 0.001. Standard error is reported for the regression intercept and coefficient, respectively.
(PDF)

## Acknowledgments

CTD station, surface PAR, and TSG data presented herein were collected by the Canadian research icebreaker *CCGS Amundsen* and made available by the Amundsen Science program, which is supported through Université Laval by the Canada Foundation for Innovation. We wish to also thank Maxim Gorbunov and Nina Schuback for valuable comments on this manuscript.

## Author Contributions

**Data curation:** Robert W. Izett, Philippe D. Tortell.

**Formal analysis:** Yayla Sezginer.

**Funding acquisition:** Philippe D. Tortell.

**Resources:** Philippe D. Tortell.

**Writing – original draft:** Yayla Sezginer.

**Writing – review & editing:** David J. Suggett, Robert W. Izett, Philippe D. Tortell.

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
