## [Decision Letter · Decision Letter 0]

14 Sep 2021

PONE-D-21-24983Chlorophyll fluorescence-based estimates of photosynthetic electron transport in Arctic phytoplankton assemblagesPLOS ONE

Dear Dr. Sezginer,

Thank you for submitting your manuscript to PLOS ONE. After careful consideration, we feel that it has merit but does not fully meet PLOS ONE’s publication criteria as it currently stands. Therefore, we invite you to submit a revised version of the manuscript that addresses the points raised during the review process.

We look forward to receiving your revised manuscript.

Kind regards,

Matheus C. Carvalho

Academic Editor

PLOS ONE

Journal Requirements:

3. In your Methods section, please provide additional location information, including geographic coordinates for the data set if available. 

5. Please update your submission to use the PLOS LaTeX template. The template and more information on our requirements for LaTeX submissions can be found at http://journals.plos.org/plosone/s/latex.

6. We note that Figures 1 and 9 in your submission contain [map/satellite] images which may be copyrighted. All PLOS content is published under the Creative Commons Attribution License (CC BY 4.0), which means that the manuscript, images, and Supporting Information files will be freely available online, and any third party is permitted to access, download, copy, distribute, and use these materials in any way, even commercially, with proper attribution. For these reasons, we cannot publish previously copyrighted maps or satellite images created using proprietary data, such as Google software (Google Maps, Street View, and Earth). For more information, see our copyright guidelines: http://journals.plos.org/plosone/s/licenses-and-copyright.

a) You may seek permission from the original copyright holder of Figures 1 and 9 to publish the content specifically under the CC BY 4.0 license.  

Reviewers' comments:

Reviewer's Responses to Questions

**Comments to the Author**

1. Is the manuscript technically sound, and do the data support the conclusions?

Reviewer #1: Partly

Reviewer #2: Yes

Reviewer #3: Yes

2. Has the statistical analysis been performed appropriately and rigorously? 

Reviewer #1: Yes

Reviewer #2: Yes

Reviewer #3: Yes

3. Have the authors made all data underlying the findings in their manuscript fully available?

Reviewer #1: Yes

Reviewer #2: Yes

Reviewer #3: Yes

4. Is the manuscript presented in an intelligible fashion and written in standard English?

Reviewer #1: Yes

Reviewer #2: Yes

Reviewer #3: Yes

5. Review Comments to the Author

Reviewer #1: Reviewer comments on manuscript entitled “Chlorophyll fluorescence-based estimates of photosynthetic electron transport in Arctic phytoplankton assemblages” authored by Yayla Sezginer, David J. Suggett, Robert W. Izett1c, and Philippe D. Tortell1 for publication in PLOS ONE.

FRRf measurements was carried out in Arctic sea, to clarify how the productivity of phytoplankton community respond to glacial and land-derived nutrients. Authors suggested a new ETRk model based on the relaxation kinetics of PSII fluorescence based on a recent study (Gorbunov & Falkowski, 2021), and compared to ETRa, which is derived by a commonly-used biophysical model. Results show that ETRa and ETRk were similar in a nutrient-rich region but clearly differed in an oligotrophic region. The sampling and FRRf measurement were carried out at a high frequency, and analysis was performed appropriately. The results are valuable data for the Arctic Ocean. I have one caveat before publication, however, and it needs careful consideration.

The major point is that there is no support of apparent (measured) O2 evolution and C assimilation rate for the two ETRs in the study area. The amplitude based ETRa have been developed as biophysical models to derive gross primary production (GPP) because FRRf can reduces only QA but not PQ pool and thus examine the electron flow rate for oxygen evolution. However, ETRa-based estimated GPP does not always equal to apparent GPP (Regaudie-de-Gioux et al., 2014). Many previous studies have investigated the factor affecting the relationships between modeled ETRa and measured O2 evolution rate (Suggett et al., 2001; Robinson et al., 2009; Deblois, Marchand & Juneau, 2013) and C assimilation rate in natural communities (Lawrenz et al., 2013; Schuback et al., 2017; Zhu et al., 2017; Hughes et al., 2018b; Ryan-Keogh et al., 2018; Kazama et al., 2021). For oxygen-based GPP, it is well confirmed for cultivated species (Suggett et al., 2009) but species composition can affect conversion factor in natural communities (Deblois et al., 2013). For, C-based GPP, although there are many studies but no global model is derived to convert ETRa to GPP (Hughes et al., 2018a). On the other hand, authors’ models is not supported by oxygen production rate or C assimilation of algae yet. The kinetic based ETRk by FIRe system (FIRe-ETRk) is a novel method to improve the errors from the parameters (Gorbunov & Falkowski, 2021). The relationships between the FIRe-ETRk and the growth rate (net primary production, NPP) of two model species, Thalassiosira pseudonana, and Dunaliella tertiolecta were examined, but not in natural community yet. Also the relationships between FIRe-ETRk and GPP is not examined yet. Therefore, the use of authors’ kinetic model in natural algal communities without reference measured productivity (O2 or C) must be considered carefully, even if it is likely analogous to the FIRe-ETRk.

Because authors’ models is not supported by oxygen production rate of algae yet, the absolute value of ETRk cannot compare to ETRa. For example, when these two ETR values are used as estimators of GPP and NPP, respectively, in the study area, the results are still questionable. Because net productivity does not include respiration, ETRk must be lower than ETRa on the same phytoplankton assemblage. However, present study clearly showed ETRk > ETRa in Lancaster Sound (Fig. 8). This paradoxical results may be due to underestimation of ETRa, and/or overestimation of ETRk. The former is likely due to inadequate dark acclimation time (5 min, P11 L211). For rapid light curve, there is no consensus rule but typically 10~20 min of dark adaptation period is used (Schuback et al., 2021).The latter is likely due to the underestimation of turnover rate, as authors pointed out. If authors primarily focus to the comparison of relative oxygen productivity between Lancaster Sound and Barrow Strait, use Fv/Fm and ETRa but not ETRk.

Minor points

P2 L25 fast repetition rate fluorometry

P5 L93 photosynthetically active radiation

P8 L157 Provide company name and city for every instruments of the manuscript.

P8 L178 Unify the unit of whole manuscript (s, min).

P10 L207 Provide the version and company of the software.

P10 L209 Table 1 What is the ChlF?

Å² PSII−1

Definition of Fq’/Fm’ (max) should not be “under 150 μmol quanta m-2 s-1”.

P11 L217 Provide the duration of each acquisitions and total time per sample.

P11 L218 Use mol instead of E.

P11 L226 Provide the duration of each light step. For rapid light curve, less than 30 s is recommended (Perkins et al., 2010).

P13 L254-258 Å² PSII−1

P13 L262 Provide the reference for 3 component multi-exponential model.

Equation (2) What are the F(t) and CQA(t)?

P13 L273-276 QA, QB

P16 L319 Provide version, company and city of the software.

P17 L349 Table 2 Correct notation Chl a.

P17 L354 Table 3 Does the zero mean true zero or lower than detection limit?

(mg m−3)

P19 L387 Spearman’s rank correlation. Use ρ (rho) or “rs” instead of R. See Schober et al. 2018. Correlation Coefficients: Appropriate Use and Interpretation, Anesthesia & Analgesia: Volume 126, 5, 1763-1768. doi: 10.1213/ANE.0000000000002864

P20 L403 Use 25th and 75th percentile, or range with median value, instead of SD.

P20 L410-415 Integrate this part into discussion.

P20 L417 Explanation is needed why 150 μmol quanta m-2 s-1 is used.

P21 L423-426 Integrate this part into discussion.

P22 L460 Fig. 8

P23 L485 Spatial variation of photophysiology?

P24 L512 Provide the reference of “model predictions”.

P24 L517 Correct notation Fv/Fm as in Table 1.

P28 L598-625 These paragraphs should be included in Methods.

P32 L683- Follow the style in journal guidelines.

Deblois C.P., Marchand A. & Juneau P. (2013). Comparison of photoacclimation in twelve freshwater photoautotrophs (chlorophyte, bacillaryophyte, cryptophyte and cyanophyte) isolated from a natural community. PLOS ONE 8, e57139. https://doi.org/10.1371/journal.pone.0057139

Gorbunov M.Y. & Falkowski P.G. (2021). Using chlorophyll fluorescence kinetics to determine photosynthesis in aquatic ecosystems. Limnology and Oceanography 66, 1–13. https://doi.org/10.1002/lno.11581

Hughes D., Campbell D., Doblin M.A., Kromkamp J., Lawrenz E., Moore C.M., et al. (2018a). Roadmaps and detours: active chlorophyll-a assessments of primary productivity across marine and freshwater systems. Environmental Science & Technology 52, 12039–12054. https://doi.org/10.1021/acs.est.8b03488

Hughes D.J., Varkey D., Doblin M.A., Ingleton T., Mcinnes A., Ralph P.J., et al. (2018b). Impact of nitrogen availability upon the electron requirement for carbon fixation in Australian coastal phytoplankton communities. Limnology and Oceanography 63, 1891–1910. https://doi.org/10.1002/lno.10814

Kazama T., Hayakawa K., Kuwahara V.S., Shimotori K., Imai A. & Komatsu K. (2021). Development of photosynthetic carbon fixation model using multi-excitation wavelength fast repetition rate fluorometry in Lake Biwa. PLOS ONE 16, e0238013. https://doi.org/10.1371/journal.pone.0238013

Lawrenz E., Silsbe G., Capuzzo E., Ylöstalo P., Forster R.M., Simis S.G.H., et al. (2013). Predicting the electron requirement for carbon fixation in seas and oceans. PLoS ONE 8, e58137. https://doi.org/10.1371/journal.pone.0058137

Perkins R.G., Kromkamp J.C., Serôdio J., Lavaud J., Jesus B., Mouget J.L., et al. (2010). The Application of Variable Chlorophyll Fluorescence to Microphytobenthic Biofilms. In: Chlorophyll a Fluorescence in Aquatic Sciences: Methods and Applications. Developments in Applied Phycology, (Eds D.J. Suggett, O. Prášil & M.A. Borowitzka), pp. 237–275. Springer Netherlands, Dordrecht.

Regaudie-de-Gioux A., Lasternas S., Agustí S. & Duarte C.M. (2014). Comparing marine primary production estimates through different methods and development of conversion equations. Frontiers in Marine Science 1, 19. https://doi.org/10.3389/fmars.2014.00019

Robinson C., Tilstone G.H., Rees A.P., Smyth T.J., Fishwick J.R., Tarran G.A., et al. (2009). Comparison of in vitro and in situ plankton production determinations. Aquatic Microbial Ecology 54, 13–34

Ryan-Keogh T.J., Thomalla S.J., Little H. & Melanson J.-R. (2018). Seasonal regulation of the coupling between photosynthetic electron transport and carbon fixation in the Southern Ocean. Limnology and Oceanography 63, 1856–1876. https://doi.org/10.1002/lno.10812

Schuback N., Hoppe C.J.M., Tremblay J.-É., Maldonado M.T. & Tortell P.D. (2017). Primary productivity and the coupling of photosynthetic electron transport and carbon fixation in the Arctic Ocean. Limnology and Oceanography 62, 898–921. https://doi.org/10.1002/lno.10475

Schuback N., Tortell P.D., Berman-Frank I., Campbell D.A., Ciotti A., Courtecuisse E., et al. (2021). Single-turnover variable chlorophyll fluorescence as a tool for assessing phytoplankton photosynthesis and primary productivity: opportunities, caveats and recommendations. Frontiers in Marine Science 0. https://doi.org/10.3389/fmars.2021.690607

Suggett D.J., Kraay G., Holligan P., Davey M., Aiken J. & Geider R. (2001). Assessment of photosynthesis in a spring cyanobacterial bloom by use of a fast repetition rate fluorometer. Limnology and Oceanography 46, 802–810. https://doi.org/10.4319/lo.2001.46.4.0802

Suggett D.J., MacIntyre H.L., Kana T.M. & Geider R.J. (2009). Comparing electron transport with gas exchange: parameterising exchange rates between alternative photosynthetic currencies for eukaryotic phytoplankton. Aquatic Microbial Ecology 56, 147–162. https://doi.org/10.3354/ame01303

Zhu Y., Ishizaka J., Tripathy S., Wang S., Sukigara C., Goes J., et al. (2017). Relationship between light, community composition and the electron requirement for carbon fixation in natural phytoplankton. Marine Ecology Progress Series 580, 83–100. https://doi.org/10.3354/meps12310

Reviewer #2: Review of PLOS ONE

Chlorophyll fluorescence-based estimates of photosynthetic electron transport in Arctic

phytoplankton assemblages Aug 29 2021

General comments

The analysis is, to my knowledge sound and robust, but I have some issues with the narrative. I think this stems to some extent to the introduction not really clearly presenting the authors’ hypotheses. I don’t think all science needs to be hypothesis-based but here we can identify several research questions but I don’t feel I got a good sense of what was anticipated and why. The end of the introduction would be where I would expect to find this, rather than the recap of key methods/results/discussion in this iteration. I would also recommend highlighting the biggest finding in the title instead of a general description of what was done. The title had me looking forwards to the “assemblages” part but since it wasn’t possible to address community composition in any way with this dataset I wonder if there isn’t a more just way to phrase this.

Another question that I would like to raise relates to the relationship between photochemical efficiency, environmental stress and species composition. In my experience, using lab cultures, fluorescence can vary a lot between taxonomic groups despite ressource replete and exponential growth conditions (notably I’d consider FvFm values to be higher for diatoms and greens and ower for (pico)cyanobacteria and heterotrophs). I guess I really feel like knowledge of species composition at lower resolution is necessary at this point to distinguish whether we are measuring stress or community changes as a result of changes in the environment (including stress!). Although I don’t think there is a whole lot of literature discussing this, I would be happy to see some support to whatever position is taken.

Finally I would have been interested in knowing how high resolution the data needs to be to have adequate dark acclimation. How many fewer samples would be analyzed with a longer dark acclimation? (10, 20, 30+ min) and how would this affect the results (maybe this adds too much to the text but comparing the statistical power of the tests used and by subsampling the current set based on different scenarioswould be one way to elegantly present this...).

If PLOS ONE does supplementary material: Do the hydrographic variables for each station need to be presented in the main text? This data is great to make available but I might stick to presenting the data directly relevant to the analyses (graphically to highlight differences between LS and BS) and leave the rest out of the main text.

Detailed comments:

Make sure the tables and figures are presented in order of appearance (I think Fig 8 is mentionned very early on). Spell check the tables.

line 318: Is it necessary to say data wasn’t corrected for autocorrelation? Otherwise specify why this wasn’t necessary.

Figures are a bit fuzzy...can this be improved?

Fig1: Should have insert of greater geographical scope to better situate sites

Fig2 and 3: grey highlight should be under curve

Fig 3: match axis colors to the colors of the curve? Improve axis clarity (put long labels on 2 lines?)

Fig4 and 5: If possible add confidence interval? Why weren’t the rank correlations calculated for BS and LS separately if we are interested in comparing these two locales

Fig 7: Fv/Fm seems related to the relationship between ETRk and ETRa...except for the steepest slope of points which has very high Fv/Fm values. This is presented in the results but isn’t discussed as far as I can tell. I would be curious to know more about why this site is different.

Fig8: hard to interpret. Could sites close by be averaged to better distinguish between day/night sites? Which could then be kept of similar size?

Reviewer #3: This is an interesting study. The article is well written and comprehensive in its interpretation of the data. With some minor revisions, detailed below, I believe it will be suitable for publication in PlosOne.

One concern I have with the paper is the use of the term “primary productivity” throughout the manuscript when speaking of the FRRf measurements. Generally (though not exclusively), this term is used to refer to either oxygen production or carbon fixation. While electron transport rates are correlated with these, they are not synonymous with them. As the authors are very well aware given their previous work, the difficulty involved in converting measurements of electron transport rates to carbon fixation has been the major stumbling block to utilizing shipboard FRRf measurements to estimate productivity in oceanographic studies. Thus, the use of the term primary production to refer to electron transport measurements derived from either of the two ETR algorithms is misleading. This is particularly so given that no effort is made in the study to directly measure photosynthetic oxygen production or carbon fixation or to determine which of the two algorithms provides a better estimate of either (as the authors point out, this is important for future work). As such, I believe it would be better to use terms like “electron transport rates” or “primary photochemistry” when referring to the measurements derived from the FRRf.

Other than that, there several minor issues within the paper that should be corrected.

Line 261: Change “ETRk” to “ETRk”.

Line 309: There are several instances in the paper where Fv/Fm is not italicized. Please make sure that italics are consistently used.

Line 394: Italicize “Fv/Fm”.

Line 460: Change “Fig8” to “Fig 8”.

Line 460: Italicize “Fv/Fm”.

Line 487: This paragraph could be cleaned up slightly. Specifically, the sentence in line 500, “we infer that nitrogen, rather than iron deficiency was the most likely cause of low photo-efficiencies”, is something of a repeat of the sentence at line 493, “suggest that the low Fv/Fm values we observed likely reflect nitrogen deficiency.”

Line 584: Change “F’q/F’m” to “Fq’/Fm’ ”.

Line 591: Change “nutrient limiting” to “nutrient-limiting”.

Line 657: Change “NQP” to “NPQ”.

Line 660: Change “F’q/F’m” to “Fq’/Fm’ ”.

Figure 3: The y-axis label for Figure E (Fq’/Fm’ 150) is somewhat confusing to read. The 150 is in line with the label for Figure D and at first it was unclear which figure it belonged to. Adjusting the figure so that the labels are not in line with or so close to each other would make it easier to read.

Figure 4: correct uE to �E in x-axis labels.

6. PLOS authors have the option to publish the peer review history of their article (what does this mean?). If published, this will include your full peer review and any attached files.

Reviewer #1: **Yes: **Takehiro Kazama

Reviewer #2: No

Reviewer #3: No

---

## [Author Response · Author response to Decision Letter 0]

17 Oct 2021

Dear Dr. Carvalho,

We wish to thank you for your consideration of this manuscript, the careful attention to detail apparent in comments from yourself and the Reviewers, and for the opportunity to revise. All requested changes (below) ultimately proved to be relatively minor, not changing the interpretation of the data or conclusion drawn. We provide our detailed point by point response as follows, providing further explanation where necessary. References to page and line numbers below correspond to the edited document containing tracked changes. We are confident that our responses address the insightful concerns raised, and have further improved our manuscript.

Yours Sincerely,

Yayla Sezginer

Responses to the Editor

1. Please ensure that your manuscript meets PLOS ONE's style requirements, including those for file naming. The PLOS ONE style templates can be found at https://journals.plos.org/plosone/s/file id=wjVg/PLOSOne_formatting_sample_main_body.pdf and https://journals.plos.org/plosone/s/file id=ba62/PLOSOne_formatting_sample_title_authors_affiliations.pdf

We have amended the manuscript to meet these style and file naming conventions.

Amundsen Science and the CCGS Amundsen were granted the following permits for our expedition:

-Nunavut Research Institute Scientific Research License # 0501119R-M

-Department of Fisheries and Oceans License to Fish for Scientific Purposes in the waters of Nunavut # S-19/20-1016-NU

-Parks Canada – National Parks Auyuittuq, Sirmilik and Quttirnipaaq # ANP-2019-32477 Canadian Wildlife Service – Access to Migratory Birds sanctuary on Bylot Island # MM-NR-2019-NU-014

-Canadian Wildlife Service – Access to National Wildlife Areas at Coburg Island, Akpait and Qaqulluit # NF-NR-2019-NU-008

-Vessel Clearance to conduct scientific work in Greenland waters. Danish Ministry of Foreign Affairs file # 2019-2308

-Permission for State Flight over Greenland File # 19/01017

-Government of Greenland Survey License # G19-036

Of these various permits, only the Nunavut Research Institute Scientific Research License #0501119R-M was required for the operations discussed in this manuscript. This license is now cited on page 7, line 180 in the Methods.

3. In your Methods section, please provide additional location information, including geographic coordinates for the data set if available.

The complete dataset of geographic coordinates where underway sampling took place is included in the salinity and temperature dataset available in the Polar Data Catalogue (doi: 10.5884/12715). A reference to the geographic coordinates in now provided in the study region map figure legend (Fig 1).

4. We note that you have stated that you will provide repository information for your data at acceptance. Should your manuscript be accepted for publication, we will hold it until you provide the relevant accession numbers or DOIs necessary to access your data. If you wish to make changes to your Data Availability statement, please describe these changes in your cover letter and we will update your Data Availability statement to reflect the information you provide

Salinity and temperature data collected by Amundsen Science are available in the Polar Data Catalogue (doi: 10.5884/12715). Surface photosynthetically active radiation data are available in the Polar Data Catalogue (doi: 10.5884/12518), alongside additional meteorological data provided by the Amundsen Science group of U. Laval. CTD station conductivity, temperature, and salinity data provided by the Amundsen Science group of U. Laval are available in the Polar Data Catalogue (doi: 10.5884/12713). Underway oxygen saturation and ΔO2/Ar data are available in the Polar Data Catalogue (doi: 10.5884/13242). All Fast Repetition Rate Fluorometry measurements of phytoplankton photophysiology have been uploaded to the Polar Data Catalogue and are currently awaiting approval. The Canadian Cryospheric Information Network reference number for this dataset is 13254, and will be trackable using this number once published. A doi will be provided as soon as available.

5. Please update your submission to use the PLOS LaTeX template. The template and more information on our requirements for LaTeX submissions can be found at http://journals.plos.org/plosone/s/latex.

We have revised the manuscript format using the PLOS LaTeX template. The reformatted manuscript and latex source code have both been uploaded under the file names Manuscript.pdf and ManuscriptSource.tex, respectively.

6. We note that Figures 1 and 9 in your submission contain [map/satellite] images which may be copyrighted. All PLOS content is published under the Creative Commons Attribution License (CC BY 4.0), which means that the manuscript, images, and Supporting Information files will be freely available online, and any third party is permitted to access, download, copy, distribute, and use these materials in any way, even commercially, with proper attribution. For these reasons, we cannot publish previously copyrighted maps or satellite images created using proprietary data, such as Google software (Google Maps, Street View, and Earth). For more information, see our copyright guidelines: http://journals.plos.org/plosone/s/licenses-and-copyright.

Both map figures were produced in Matlab R2020a using the freely available m_map package available online at www.eoas.ubc.ca/~rich/map.html. The package includes a coastline, global elevation database, and river database, and provides access to publicly available bathymetry data. The data are not proprietary. A citation for m_map has been added to the map figure legend and the references section (Fig 1).

Responses to Reviewer 1:

“The major point is that there is no support of apparent (measured) O2 evolution and C assimilation rate for the two ETRs in the study area. The amplitude based ETRa have been developed as biophysical models to derive gross primary production (GPP) because FRRf can reduces only QA but not PQ pool and thus examine the electron flow rate for oxygen evolution. However, ETRa-based estimated GPP does not always equal to apparent GPP (Regaudie-de- Gioux et al., 2014). Many previous studies have investigated the factor affecting the relationships between modeled ETRa and measured O2 evolution rate (Suggett et al., 2001; Robinson et al., 2009; Deblois, Marchand & Juneau, 2013) and C assimilation rate in natural communities (Lawrenz et al., 2013; Schuback et al., 2017; Zhu et al., 2017; Hughes et al., 2018b; Ryan-Keogh et al., 2018; Kazama et al., 2021). For oxygen-based GPP, it is well confirmed for cultivated species (Suggett et al., 2009) but species composition can affect conversion factor in natural communities (Deblois et al., 2013). For, C-based GPP, although there are many studies but no global model is derived to convert ETRa to GPP (Hughes et al., 2018a). On the other hand, authors’ models is not supported by oxygen production rate or C assimilation of algae yet. The kinetic based ETRk by FIRe system (FIRe-ETRk) is a novel method to improve the errors from the parameters (Gorbunov & Falkowski, 2021). The relationships between the FIRe-ETRk and the growth rate (net primary production, NPP) of two model species, Thalassiosira pseudonana, and Dunaliella tertiolecta were examined, but not in natural community yet. Also the relationships between FIRe-ETRk and GPP is not examined yet. Therefore, the use of authors’ kinetic model in natural algal communities without reference measured productivity (O2 or C) must be considered carefully, even if it is likely analogous to the FIRe-ETRk. Because authors’ models is not supported by oxygen production rate of algae yet, the absolute value of ETRk cannot compare to ETRa. For example, when these two ETR values are used as estimators of GPP and NPP, respectively, in the study area, the results are still questionable. Because net productivity does not include respiration, ETRk must be lower than ETRa on the same phytoplankton assemblage. However, present study clearly showed ETRk > ETRa in Lancaster Sound (Fig. 8). This paradoxical results may be due to underestimation of ETRa, and/or overestimation of ETRk. The former is likely due to inadequate dark acclimation time (5 min, P11 L211). For rapid light curve, there is no consensus rule but typically 10~20 min of dark adaptation period is used (Schuback et al., 2021).The latter is likely due to the underestimation of turnover rate, as authors pointed out. If authors primarily focus to the comparison of relative oxygen productivity between Lancaster Sound and Barrow Strait, use Fv/Fm and ETRa but not ETRk.”

The Reviewer makes excellent points but appears to have mis-interpreted the aims and scope of our study. We agree that electron transport rates (ETR) are not equivalent to oxygen evolution or carbon uptake. Indeed, as noted by the reviewer, there are many photosynthetic and metabolic processes that can lead to the decoupling of electron transport from oxygen evolution and downstream carbon assimilation. However, it was not our intention in this article to use ETR as an absolute metric of primary productivity. Rather, as we state in the introduction (P6 L150), our goal is to compare different approaches to estimating ETR, examining how the two contrasting algorithms perform across a range of hydrographic regimes. As pointed out by the Reviewer, there is a significant need for work of this kind in natural environments, moving beyond current results that have focused on laboratory studies. Even without ‘calibration’ against independent productivity bench-marks (O2 evolution, CO2 uptake), ETR is an important diagnostic of photosynthetic capacity in its own right. The reducing power produced by this process fuels a number of metabolic functions, and presents an upper limit for carbon fixation by the Calvin-Benson cycle. We understand that the comments of the reviewer may have reflected a lack of clarity in our initial wording. We therefore have altered our wording throughout from ‘primary productivity’ to ‘primary photochemistry’ to remove any suggestion that ETR is interchangeable with carbon uptake or oxygen evolution. We wholeheartedly agree that validating ETR measurements with parallel measures of O2 evolution would enable us to assess the accuracy of the different ETR algorithms. This has now been further emphasized in the discussion section, where we suggest such work for a follow up study (P3 L862). The reviewer also raises some concerns about the effects of respiration on our results, suggesting that ETRk and ETRa measure NPP and GPP, respectively. This is, in fact, incorrect, as both ETR algorithms provide estimates of GPP. The kinetic ETR algorithm recently developed by Gorbunov and Falkowski (2021) evaluates the rate of electron transport through Photosystem II (PSII) as the normalized photochemical yield at a given light intensity multiplied by the rate at which the primary electron acceptor, Qa is capable of turning over electrons from the photosynthetic Reaction Center II to the secondary electron acceptor, Qb. As such, ETRk measures primary photochemistry and is theoretically equivalent to ETRa, which measures the rate of electron transport through PS II as the product of the photochemical energy conversion efficiency, the functional light absorption area of PSII, and the provided light intensity. Although Gorbunov and Falkowski (2021) used the high correlation between lab-grown phytoplankton

growth rates and ETRk to validate their novel kinetic-based ETR algorithm, ETRk is not a measure of NPP, which is the difference between GPP and respiration.

Minor points

P2 L25 fast repetition rate fluorometry

->No change, we prefer to keep the capitalization for consistency with the literature.

P5 L93 photosynthetically active radiation

->Resolved. (P5 L106)

P8 L157 Provide company name and city for every instruments of the manuscript.

->Provided company name where missing. Did not include city name, to keep with style of

previous PLOS One publications (see Schuback et al., 2015)

P8 L178 Unify the unit of whole manuscript (s, min).

->No change (unless the Editor and Editorial process requires) since our results capture processes

occurring on differing time-scales, we prefer to use different units of time, as appropriate.

P10 L207 Provide the version and company of the software.

->Resolved company name. Software version/name provided.

P10 L209 Table 1 What is the ChlF?

->Replaced with Chla

Å² PSII−1, Definition of Fq’/Fm’ (max) should not be “under 150 μmol quanta m-2 s-1”.

->Resolved.

P11 L217 Provide the duration of each acquisitions and total time per sample.

->Resolved. P10 L253

P11 L218 Use mol instead of E.

->Resolved.

P11 L226 Provide the duration of each light step. For rapid light curve, less than 30 s is

recommended (Perkins et al., 2010).

->Resolved. P10 L260

P13 L254-258 Å² PSII−1

->Resolved.

P13 L262 Provide the reference for 3 component multi-exponential model.

->Resolved. P13 L325

Equation (2) What are the F(t) and CQA(t)?

->Resolved. Measured fluorescence and fraction of RCIIs closed by excitation flashlets. P14 L345

P13 L273-276 QA, QB

->Set to Qa and Qb throughout, removed all instances of QA and QB

P16 L319 Provide version, company and city of the software.

->Added company. Version included. (P13 L342)

P17 L349 Table 2 Correct notation Chl a.

->Resolved. Table 2 moved to the supplements.

P17 L354 Table 3 Does the zero mean true zero or lower than detection limit?

->Zeros do indicate below detection limit. Values below detection limit in have been updated to <

0.02 uM. Table 3 has been moved to the Supporting Information section (S2 Table)

P19 L387 Spearman’s rank correlation. Use ρ (rho) or “rs” instead of R. See Schober et al. 2018.

Correlation Coefficients: Appropriate Use and Interpretation, Anesthesia & Analgesia: Volume

126, 5, 1763-1768. doi: 10.1213/ANE.0000000000002864

->Resolved throughout.

P20 L403 Use 25th and 75th percentile, or range with median value, instead of SD.

->Deviation from the median reported is the median absolute deviation, not the SD.

P20 L410-415 Integrate this part into discussion.

->We prefer to retain this aspect in the results, as these lines provide a quick operational note,

rather than a discussion of key findings.

P20 L417 Explanation is needed why 150 μmol quanta m-2 s-1 is used.

->Explanation added to Methods (P11 L283)

P21 L423-426 Integrate this part into discussion.

->Agreed. We have moved the majority of text outlining our interpretations of Fig 5 to P24 L672-

678 in the Discussion section. 

P22 L460 Fig. 8

->Resolved

P23 L485 Spatial variation of photophysiology?

->Adopted suggestion. (P22 L609)

P24 L512 Provide the reference of “model predictions”.

->Resolved. (P23 L637)

P24 L517 Correct notation Fv/Fm as in Table 1.

->Resolved throughout.

P28 L598-625 These paragraphs should be included in Methods

->After some consideration, we are confident that these paragraphs belong in the discussion, since

they provide critical context to understand how differing numerical procedures may affect

results. Further, these lines include a description of an analogous method that we did not apply -

including this in the Methods could lead to some confusion in what we actually do, as opposed to

what we contrast against.

P32 L683- Follow the style in journal guidelines.

->Resolved as per Editor’s comments.

Responses to Reviewer 2:

“The analysis is, to my knowledge sound and robust, but I have some issues with the narrative. I think this stems to some extent to the introduction not really clearly presenting the authors’ hypotheses. I don’t think all science needs to be hypothesis-based but here we can identify several research questions but I don’t feel I got a good sense of what was anticipated and why. The end of the introduction would be where I would expect to find this, rather than the recap of key methods/results/discussion in this iteration. I would also recommend highlighting the biggest finding in the title instead of a general description of what was done. The title had me looking forwards to the “assemblages” part but since it wasn’t possible to address community composition in any way with this dataset I wonder if there isn’t a more just way to phrase this. Another question that I would like to raise relates to the relationship between photochemical efficiency, environmental stress and species composition. In my experience, using lab cultures, fluorescence can vary a lot between taxonomic groups despite ressource replete and exponential growth conditions (notably I’d consider FvFm values to be higher for diatoms and greens and ower for (pico)cyanobacteria and heterotrophs). I guess I really feel like knowledge of species composition at lower resolution is necessary at this point to distinguish whether we are measuring stress or community changes as a result of changes in the environment (including stress!). Although I don’t think there is a whole lot of literature discussing this, I would be happy to see some support to whatever position is taken. Finally I would have been interested in knowing how high resolution the data needs to be to have adequate dark acclimation. How many fewer samples would be analyzed with a longer dark

acclimation? (10, 20, 30+ min) and how would this affect the results (maybe this adds too much to the text but comparing the statistical power of the tests used and by subsampling the current set based on different scenarios would be one way to elegantly present this...). If PLOS ONE does supplementary material: Do the hydrographic variables for each station need to be presented in the main text? This data is great to make available but I might stick to presenting the data directly relevant to the analyses (graphically to highlight differences between LS and BS) and leave the rest out of the main text.”

The Reviewer raises important points that we have addressed:

1. We have added our research question and objective to the introductory paragraph (P6 L150), and refined our title, as suggested, to ensure more transparency in what was anticipated and why.

2. We agree entirely with the Reviewers point, and have shown previously from broad analysis of numerous cultures (e.g. Suggett et al. 2009) that diatoms and greens result in very different inherent FRRf parameterization (e.g. higher Fv/Fm) than pico and nano phytoplankton, but also that these trends are hard to deconvolve from patterns of resource availability for photosynthesis. We therefore argue that variability in Fv/Fm results from both direct physiological effects of nutrient limitation, and indirect effects that increase the abundance of smaller phytoplankton taxa (with typically lower Fv/Fm) in nutrient limited environments. We make this point on P24 L680-714, and acknowledge the unfortunate lack of species composition data in our study to fully resolve this issue.

3. We recently conducted additional experiments to address the question of dark acclimation time-scales. (These experiments were conducted this past year, i.e. on a different cruise from that presented in this paper). We allowed samples to dark acclimate over 30 minutes and took FRRf measurements every 2 minutes to assess Fv/Fm and 𝜎!"## recovery throughout the course of the dark acclimation period. Samples required a mean recovery time of 9.65 ± 1.53 min (n = 34) to reach stable, maximum light absorption efficiencies. The sampling approach used in this current study (2019 data) used a 5 min acclimation time. Increasing this to 10 minutes would increase the overall sampling interval to 17 minutes, which would have reduced our sample size from 481 to 340, representing a 29.5% reduction in measurement resolution. These preliminary results have been added to the Discussion section (P26 L749-764).

4. We prefer to keep the hydrographic data in the main text to provide context for the physical properties of our sampling region. We have moved the table of correlation statistics between hydrographic variables to the supplemental section, as we agree that it is not critical to the narrative presented. All the hydrographic data is publicly available on the Polar Data Catalogue (doi: 10.5884/12715).

Detailed comments:

Make sure the tables and figures are presented in order of appearance (I think Fig 8 is mentioned very early on). Spell check the tables.

->Resolved.

line 318: Is it necessary to say data wasn’t corrected for autocorrelation? Otherwise specify why this wasn’t necessary.

->We have removed this line.

Figures are a bit fuzzy...can this be improved?

->High resolution versions of the figures can be downloaded by clicking the link at the top right corner of the figure page.

Fig1: Should have insert of greater geographical scope to better situate sites

->Resolved.

Fig2 and 3: grey highlight should be under curve

->Resolved.

Fig 3: match axis colors to the colors of the curve? Improve axis clarity (put long labels on 2 lines?)

->Resolved.

Fig4 and 5: If possible add confidence interval? Why weren’t the rank correlations calculated for BS and LS separately if we are interested in comparing these two locales

->Resolved. Computing separate correlations for BS and LS has led to some new discussion of the differences between these two regions, particularly regarding Fq’/Fm’(150) and PAR relationship. We interpret the positive relationship between Fq’/Fm’ (150) and PAR in Barrow Strait as an indicator of the increased ability of phytoplankton in Barrow Strait to acclimate to higher light levels. In contrast, the lack of a relationship between Fq’/Fm’ (150) and PAR in Lancaster Sound is further evidence of possible photodamage to reaction centers, likely due to insufficient N

required to repair non-functional reaction centers. We discuss these results on P24 L672-678.

Fig 7: Fv/Fm seems related to the relationship between ETRk and ETRa...except for the steepest

slope of points which has very high Fv/Fm values. This is presented in the results but isn’t

discussed as far as I can tell. I would be curious to know more about why this site is different.

->The unusual site where Fv/Fm is very high, but has a large decoupling ratio between ETRk:ETRa is unique in that it has a very low Fq’/Fm’emax value ( < 0.04) despite its high Fv/Fm value ( > 0.42). Fv/Fm and Fq’/Fm’emax are highly correlated (𝜌 = 0.93 in our dataset), so it is likely this site is an outlier given we have one of our highest Fv/Fm and lowest Fq’/Fm’emax values here. This explanation is now given on P27 L792-809.

Fig8: hard to interpret. Could sites close by be averaged to better distinguish between day/night sites? Which could then be kept of similar size?

->After spending some time rethinking how to best represent the data presented in Fig 9 (we believe Reviewer 2 is actually commenting on Fig 9), we’ve resolved to keep the figure as is. We believe that averaging samples across broad areas would obscure small-scale features in photophysiological spatial variability, which we believe is one of the unique and valuable aspects of our study.

Responses to Reviewer 3:

“One concern I have with the paper is the use of the term “primary productivity” throughout the manuscript when speaking of the FRRf measurements. Generally (though not exclusively), this term is used to refer to either oxygen production or carbon fixation. While electron transport rates are correlated with these, they are not synonymous with them. As the authors are very well aware given their previous work, the difficulty involved in converting measurements of electron transport rates to carbon fixation has been the major stumbling block to utilizing shipboard FRRf measurements to estimate productivity in oceanographic studies. Thus, the use of the term

primary production to refer to electron transport measurements derived from either of the two ETR algorithms is misleading. This is particularly so given that no effort is made in the study to directly measure photosynthetic oxygen production or carbon fixation or to determine which of the two algorithms provides a better estimate of either (as the authors point out, this is important for future work). As such, I believe it would be better to use terms like “electron transport rates” or “primary photochemistry” when referring to the measurements derived from the FRRf.”

In agreement also with Reviewer 1’s first comment, we have adopted the Reviewer 3’s suggestion of switching our language from ‘primary productivity’ to ‘primary photochemistry’ to avoid confusion with carbon or oxygen-based productivity measures.

Line 261: Change “ETRk” to “ETRk”.

->Resolved.

Line 309: There are several instances in the paper where Fv/Fm is not italicized. Please make sure that italics are consistently used.

->Resolved.

Line 460: Change “Fig8” to “Fig 8”.

->Resolved.

Line 487: This paragraph could be cleaned up slightly. Specifically, the sentence in line 500, “we infer that nitrogen, rather than iron deficiency was the most likely cause of low photoefficiencies”, is something of a repeat of the sentence at line 493, “suggest that the low Fv/Fm values we observed likely reflect nitrogen deficiency.”

->Removed repetitive statements.

Line 584: Change “F’q/F’m” to “Fq’/Fm’ ”.

->Resolved

Line 591: Change “nutrient limiting” to “nutrient-limiting”.

->Resolved.

Line 657: Change “NQP” to “NPQ”.

->Resolved.

Line 660: Change “F’q/F’m” to “Fq’/Fm’ ”.

->Resolved.

Figure 3: The y-axis label for Figure E (Fq’/Fm’ 150) is somewhat confusing to read. The 150 is in line with the label for Figure D and at first it was unclear which figure it belonged to. Adjusting the figure so that the labels are not in line with or so close to each other would make it easier to read.

->Resolved.

Figure 4: correct uE to μE in x-axis labels.

->Resolved.

Once again, we thank each of the Reviewers for their insightful questions and comments, which we feel have enhanced the quality of this manuscript.

---

## [Decision Letter · Decision Letter 1]

17 Nov 2021

Irradiance and nutrient-dependent effects on photosynthetic electron transport in Arctic phytoplankton: a comparison of two Chlorophyll fluorescence-based approaches to derive primary photochemistry

PONE-D-21-24983R1

Dear Dr. Sezginer,

We’re pleased to inform you that your manuscript has been judged scientifically suitable for publication and will be formally accepted for publication once it meets all outstanding technical requirements.

Kind regards,

Matheus C. Carvalho

Academic Editor

PLOS ONE

Additional Editor Comments (optional):

Reviewers' comments:

Reviewer's Responses to Questions

**Comments to the Author**

1. If the authors have adequately addressed your comments raised in a previous round of review and you feel that this manuscript is now acceptable for publication, you may indicate that here to bypass the “Comments to the Author” section, enter your conflict of interest statement in the “Confidential to Editor” section, and submit your "Accept" recommendation.

Reviewer #1: All comments have been addressed

Reviewer #2: All comments have been addressed

Reviewer #3: All comments have been addressed

2. Is the manuscript technically sound, and do the data support the conclusions?

Reviewer #1: Yes

Reviewer #2: Yes

Reviewer #3: (No Response)

3. Has the statistical analysis been performed appropriately and rigorously? 

Reviewer #1: Yes

Reviewer #2: Yes

Reviewer #3: (No Response)

4. Have the authors made all data underlying the findings in their manuscript fully available?

Reviewer #1: Yes

Reviewer #2: Yes

Reviewer #3: (No Response)

5. Is the manuscript presented in an intelligible fashion and written in standard English?

Reviewer #1: Yes

Reviewer #2: Yes

Reviewer #3: (No Response)

6. Review Comments to the Author

Reviewer #1: The new title is explaining the concept of this paper concisely. The issue of the uncertainty in the relationship between ETR and productivity has been solved by shifting to “primary photochemistry” instead of “primary productivity”. The methods and results are well-documented. I think this revised manuscript is good to publish after fixing some minor points such as follows.

Minor points:

Title: chlorophyll

Table 1 and main body: The term p can be easily confused with the p of the p-value. Please change either of these.

L183: 30 s

Table 2: Use the minus sign despite the hyphen.

L420: CO2

L840-L851: CQa(t)

Reviewer #2: My concerns with the manuscript have been adequately addressed. I might suggest another mention of the potential role of taxonomy to better unravel the relationship between photophysiology, fluorescence and productivity in the conclusion section but this is potentially more indicative of my particular biases.

Reviewer #3: (No Response)

7. PLOS authors have the option to publish the peer review history of their article (what does this mean?). If published, this will include your full peer review and any attached files.

Reviewer #1: **Yes: **Takehiro Kazama

Reviewer #2: No

Reviewer #3: No

---

## [Editor Report · Acceptance letter]

1 Dec 2021

PONE-D-21-24983R1 

Irradiance and nutrient-dependent effects on photosynthetic electron transport in Arctic phytoplankton: a comparison of two Chlorophyll fluorescence-based approaches to derive primary photochemistry 

Dear Dr. Sezginer:

I'm pleased to inform you that your manuscript has been deemed suitable for publication in PLOS ONE. Congratulations! Your manuscript is now with our production department. 

Kind regards, 

on behalf of

Dr. Matheus C. Carvalho 

Academic Editor

PLOS ONE